# A polycyclic scaffold identified by structure-based drug design effectively inhibits the human P2X7 receptor

Adam C. Oken [1], Andreea L. Turcu [2,3], Eva Tzortzini[4], Kyriakos Georgiou[4], Jessica Nagel [5], Franka G. Westermann [5], Marta Barniol-Xicota[2,9], Jonas Seidler[6], Ga-Ram Kim[7], So-Deok Lee[7], Annette Nicke [6], Yong-Chul Kim[7], Christa E. Müller [5], Antonios Kolocouris [4], Santiago Vázquez [2,3] & Steven E. Mansoor [1,8] ✉

The P2X7 receptor is an ATP-gated ion channel that activates inflammatory pathways involved in diseases such as cancer, atherosclerosis, and neurodegeneration. However, despite the potential benefits of blocking overactive signaling, no P2X7 receptor antagonists have been approved for clinical use. Understanding species-specific pharmacological effects of existing antagonists has been challenging, in part due to the dearth of molecular information on receptor orthologs. Here, to identify distinct molecular features in the human receptor, we determine high-resolution cryo-EM structures of the full-length wild-type human P2X7 receptor in apo closed and ATP-bound open state conformations and draw comparisons with structures of other orthologs. We also report a cryo-EM structure of the human receptor in complex with an adamantane-based inhibitor, which we leverage, in conjunction with functional data and molecular dynamics simulations, to design a potent and selective antagonist with a unique polycyclic scaffold. Functional and structural analysis reveal how this optimized ligand, termed UB-MBX-46, interacts with the classical allosteric pocket of the human P2X7 receptor with sub-nanomolar potency and high selectivity, revealing its significant therapeutic potential.

When present in high concentrations, such as in pathological inflammatory states, extracellular ATP acts as a danger signal to cells by binding to and activating the P2X7 receptor (P2X7R)[1–5]. This spatiotemporal signaling stimulates the innate immune system by triggering assembly of the NLRP3 inflammasome and subsequent cytokine release, as well as activation of various less understood signaling cascades, ultimately resulting in apoptosis[6–12]. Modulation of P2X7R-mediated cellular responses has the potential to treat diseases such as

[1]Department of Chemical Physiology & Biochemistry, Oregon Health & Science University, Portland, OR, USA. [2]Laboratori de Química Farmacèutica, Facultat de Farmàcia i Ciències de l'Alimentació, Universitat de Barcelona, Barcelona, Spain. [3]Institute of Biomedicine of the University of Barcelona, IBUB, Barcelona, Spain. [4]Laboratory of Medicinal Chemistry, Section of Pharmaceutical Chemistry, Department of Pharmacy, National and Kapodistrian University of Athens, Panepistimiopolis-Zografou, Greece. [5]PharmaCenter Bonn & Pharmaceutical Institute, Pharmaceutical & Medicinal Chemistry, University of Bonn, Bonn, Germany. [6]Walther Straub Institute of Pharmacology and Toxicology, Faculty of Medicine, Ludwig-Maximilians-Universität München, Munich, Germany. [7]School of Life Sciences, Gwangju Institute of Science and Technology, Buk-gu, Gwangju, Republic of Korea. [8]Division of Cardiovascular Medicine, Knight Cardiovascular Institute, Oregon Health & Science University, Portland, OR, USA. [9]Present address: Department of Medicine and Life Sciences, Biomedical Research Park (PRBB), Universitat Pompeu Fabra, Barcelona, Spain. ✉e-mail: mansoors@ohsu.edu

atherosclerosis in the cardiovascular system, Alzheimer's disease in the central nervous system, as well as autoimmune diseases, infections, and select cancers in the immune system[4,13–21]. For these reasons, the P2X7R is a particularly important drug target.

While only few antagonists target the orthosteric binding site, there are numerous chemically distinct allosteric P2X7R antagonists, with wide-ranging potencies across orthologs. All antagonists discussed throughout the rest of this manuscript are allosteric antagonists. These ligands feature scaffolds containing adamantane or heterocycles such as tetrazole, pyridine, or quinoline (Supplementary Fig. 1)[22–24]. Although several P2X7R antagonists have progressed to clinical trials and safety has been well established, the outcomes have been disappointing, and none have reached the market[25,26]. This could be partially attributed to the species-specific P2X7R expression patterns and variable pharmacological effects that each antagonist has on P2X7Rs from different species, complicating the predictability and translation of animal data to human efficacy. For instance, the P2X7R antagonist AZD9056, which reached phase II trials for rheumatoid arthritis (RA), ultimately showed limited efficacy, highlighting the challenges for extrapolating data from animal models to human outcomes[25]. Other examples of P2X7R antagonists that have shown variations in activity across orthologs include JNJ47965567, which has ~10-fold higher affinity for the rat P2X7R (rP2X7R) than the human P2X7R (hP2X7R) (Supplementary Fig. 1)[27]. In contrast, AZ11645373 is ~100-fold less potent at the mouse P2X7R (mP2X7R), and >500-fold less effective against the rP2X7R, than the hP2X7R (Supplementary Fig. 1)[28,29]. There are also P2X7R antagonists, such as A438079, that have equal inhibitory activity across all three orthologs but are generally less potent compared to more ortholog-specific antagonists (Supplementary Fig. 1)[5,30].

As a result, there is a therapeutic need for small molecules that specifically and potently modulate hP2X7Rs while maintaining sufficient potency in rodents required for preclinical studies. Structure-based drug design could meet this need, but our poor molecular understanding of the pharmacological variability between P2X7R orthologs has delayed progress[25,31,32]. Structural investigations of the truncated panda P2X7R (pdP2X7R) and the full-length rP2X7R provided initial insight into the mechanism of allosteric antagonism of P2X7Rs[33,34]. P2X7R antagonists bind to either classical or extended allosteric ligand-binding sites that exist within the extracellular domain at the interface of two protomers[33,34]. Such antagonists have been classified into shallow, deep, or starfish binders, depending on their functional properties and the residues within distinct allosteric ligand-binding sites with which they interact[34]. For example, JNJ47965567 is a deep-binding ligand that occupies the classical allosteric binding site, whereas methyl blue is a starfish-binding ligand that occupies the extended allosteric binding site (Supplementary Figs. 1 and 2)[34]. Although these data define the overarching principles of allosteric antagonism for the P2X7R, a comprehensive description of the ligand-binding sites and the associated molecular pharmacology that distinguishes different P2X7R orthologs remains enigmatic.

Here, to guide the development of hP2X7R ligands for therapeutic intervention, we use structural, biophysical, and electrophysiological methods to describe the molecular pharmacology of full-length wild-type rat, mouse, and human P2X7Rs. Our cryo-EM structures of the mP2X7R and the hP2X7R in the apo closed state, and the hP2X7R in the ATP-bound open state, identify ortholog-specific conformational differences and previously uncharacterized cholesterol binding sites. Further, through comparison with our previously published structure of the rP2X7R, we reveal differences in the classical allosteric ligand-binding site that underlie the pharmacological diversity between these orthologs[35,36]. We also determine the cryo-EM structure of the adamantane-based inhibitor UB-ALT-P30 bound to the hP2X7R, revealing the potential for larger scaffold ligands to bind within the classical allosteric pocket and leverage structure-based drug design to develop a potent and specific polycyclic scaffold for the hP2X7R. With this knowledge, we design and synthesize several antagonist scaffolds and identify a promising compound, UB-MBX-46, for functional and cryo-EM analysis. UB-MBX-46 binds to the hP2X7R with subnanomolar potency and high selectivity and thus defines a ligand scaffold with the potential to treat P2X7R-associated diseases.

## Results

### P2X7R orthologs have distinct allosteric binding sites

To characterize the molecular differences between key mammalian P2X7R orthologs, we obtained cryo-EM structures of the full-length wild-type mP2X7 (2.5 Å) and hP2X7 (2.5 Å) receptors in the apo closed state conformation (Fig. 1A, Supplementary Figs. 3 and 4 and Supplementary Table 1). The overall architecture of this P2X receptor (P2XR) subtype is consistent across human, mouse, and rat (PDB: 8TR5) orthologs (mean RMSD = 0.7 Å between Cα carbons)[35]. Conserved features include their trimeric architecture, the $3_{10}$-helices forming a closed gate, a partially hydrated sodium ion present in the closed pore, palmitoylation of residues in the C-cys anchor, and zinc and guanosine nucleotide-binding sites in the cytoplasmic ballast (Fig. 1 and Supplementary Fig. 2A, 5)[35,36]. However, we also identified ortholog-specific features in the hP2X7R, including two cholesterol hemisuccinate (CHS) binding sites per protomer at the interface between transmembrane helix 1 (TM1) and transmembrane helix 2 (TM2) on what would be the extracellular leaflet of the membrane bilayer (Fig. 1B–D). The two CHS molecules are stacked on top of each other such that the inner molecule is closer to the TM1/TM2 interface, and the outer molecule is positioned between TM1 and the inner molecule (Fig. 1C, D). The inner CHS molecule is coordinated by hydrophobic interactions with residues F328 and L333 on TM2; F38, V41, C42, L45, and Y51 on TM1; and W265 on the lower body domain (Fig. 1C and Supplementary Fig. 2A). Two oxygen atoms on the hemisuccinate group of the inner CHS molecule also form hydrogen bonds with the side chain of R264 and backbone nitrogen of W265 (distances of 2.5 Å and 2.9 Å, respectively) (Fig. 1C). The outer CHS molecule is coordinated by hydrophobic interactions with residues F38, C42, and V46 on TM1; residue F266 on the lower body domain; and the inner CHS molecule (Fig. 1D and Supplementary Fig. 2A). There are also hydrogen bonds between the hemisuccinate group of the outer CHS molecule and the side chains of N261, H268, and K49 (distances of 2.8 Å, 3.4 Å, and 3.3 Å, respectively) (Fig. 1D). Although all three ortholog reconstructions are of similar resolution (~2.5 Å), there are no densities in either the rP2X7R or the mP2X7R reconstructions to support modeling of CHS molecules, suggesting that these bound CHS molecules are specific to the hP2X7R.

Comparison of these three structures also reveals molecular differences in the unoccupied classical allosteric ligand-binding sites (Fig. 1A, E–G)[34,35]. Across the three orthologs, the classical allosteric pocket comprises multiple conserved residues, including F88, D92, F103, M105, K110, F293, K297, Y295, K297, Y298, E305, and I310 (Fig. 1E–G and Supplementary Fig. 6). However, the identity of three residues within the classical allosteric pocket are different between human, mouse, and rat orthologs as seen in sequence alignments and visualized in their respective structures (Fig. 1E–G and Supplementary Figs. 6 and 7). First, F95 in hP2X7Rs and mP2X7Rs correlates to L95 in rP2X7Rs (Fig. 1E–G Supplementary Figs. 6 and 7). This residue occupies the same general location and rotameric orientation in all three orthologs, but the phenylalanine residue found in human and mouse P2X7Rs occupies much more space within the classical allosteric pocket[35]. Second, F108 in hP2X7Rs correlates to Y108 in both mP2X7Rs and rP2X7Rs (Fig. 1E–G and Supplementary Figs. 6 and 7). Again, this residue occupies the same position and rotameric conformation in all three structures, likely playing a similar role (Fig. 1E–G)[35]. Finally, V312 in hP2X7Rs correlates to A312 in both mP2X7Rs and rP2X7Rs (Fig. 1E–G and Supplementary Figs. 6 and 7). The larger valine side chain in the

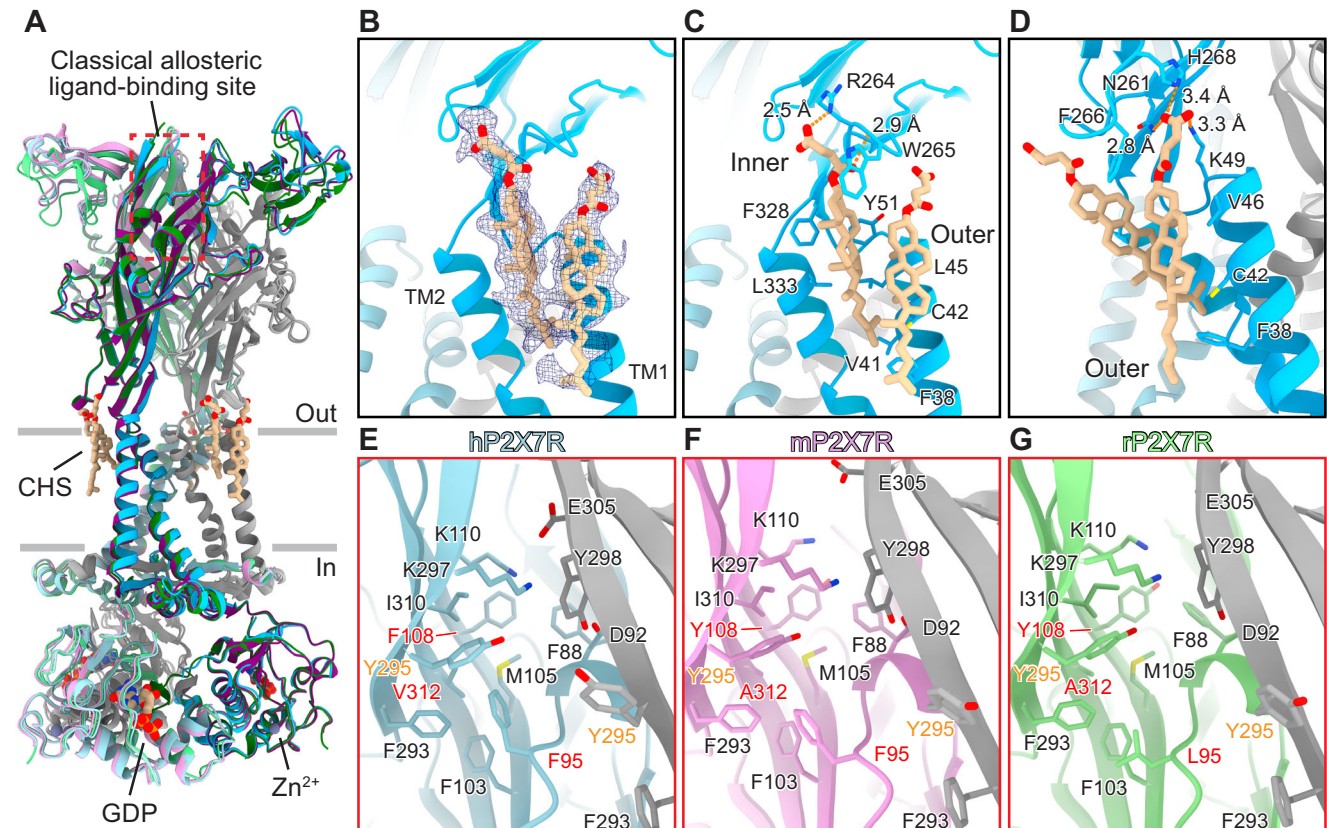

**Fig. 1 | Structures of human, mouse, and rat P2X7Rs in the apo closed state conformation reveal distinct allosteric ligand-binding sites. A** Ribbon representation of the apo closed state structures of human (shades of blue and gray), mouse (shades of purple and gray), and rat (shades of green and gray, PDB code: 8TR5) P2X7Rs, aligned with ChimeraX[35]. The classical allosteric ligand-binding site is boxed in red. Two CHS molecules per protomer (inner and outer; both tan), located at the interface between TM1 and TM2 on the extracellular side of the membrane, are found only in the hP2X7R. GDP (tan) and Zn²⁺ ions (slate gray) are labeled and shown within the cytoplasmic ballast (Supplementary Fig. 2 A). **B** Magnified and 120° rotated view of **A**, highlighting cryo-EM density for two molecules of CHS bound to one protomer of the hP2X7R. **C** Same view as

**B**, highlighting the hydrophobic and hydrogen bonding interactions between the inner CHS molecule and the hP2X7R. **D** 50° rotated view from **C** highlighting the hydrophobic and hydrogen bonding interactions between the outer CHS molecule and the hP2X7R. **E-G** Magnified view of the classical allosteric ligand-binding site from **A**, highlighting the differences between human (**E**, light blue and gray), mouse (**F**, light purple and gray), and rat (**G**, light green and gray) P2X7R orthologs[35]. Residues 95, 108, and 312 (red labels) are the key residue differences in the classical allosteric pockets between the three orthologs. The larger V312 in hP2X7Rs (A312 in mP2X7Rs and rP2X7Rs) forces the neighboring residue Y295 (orange labels) to adopt an alternative rotameric conformation that condenses the classical allosteric pocket in the human ortholog only (Supplementary Fig. 7).

human ortholog occupies more space within the classical allosteric pocket than the alanine in rat or mouse P2X7Rs (Fig. 1E–G and Supplementary Figs. 6 and 7)[35]. Interestingly, although not apparent from sequence alignments, our structure of the hP2X7R shows that the larger side chain of V312 forces the adjacent, conserved residue Y295 to adopt an alternative rotameric conformation compared to its conformation in both mP2X7R and rP2X7R, condensing the size of the pocket in hP2X7R (Fig. 1E–G and Supplementary Fig. 6 and 7). Thus, the classical allosteric ligand-binding site in the human ortholog has a different size and shape, which impacts the binding of allosteric antagonists.

**P2X7R orthologs have distinct orthosteric ATP-binding sites**

Because pharmacological tools that activate human, mouse, and rat P2X7Rs are distinct, we examined the molecular determinants of agonism in the human ortholog to facilitate the development of higher-affinity agonists[30,35–37]. In agreement with previous measurements, two-electrode voltage clamp (TEVC) recordings of human, mouse, and rat P2X7Rs in the absence of divalent cations revealed half maximal effective concentrations (EC₅₀) for ATP of $89 \pm 8.3\,\mu M$, $70 \pm 17\,\mu M$, and $34 \pm 8.4\,\mu M$, respectively (Fig. 2A and Supplementary Fig. 8)[35,36,38]. The modestly lower apparent affinity of ATP for the hP2X7R was borne out by direct measurements of kinetics and

equilibrium binding affinities using bio-layer interferometry (BLI). The rate constant for association ($k_a$) of $7.4 \pm 1.3 \times 10^4\,M^{-1}\,s^{-1}$, and rate constant for dissociation ($k_d$) of $4.8 \pm 1.2 \times 10^{-2}\,s^{-1}$, are ~27% and ~6% slower than those for ATP binding to the rP2X7R (Fig. 2B)[35]. This corresponds to ATP binding to the hP2X7R with an equilibrium dissociation constant ($K_D = 650 \pm 120\,nM$) that is ~20% higher the rP2X7R ($K_D = 540 \pm 230\,nM$) (Fig. 2B)[35].

To gain molecular insight into the binding of ATP to hP2X7Rs, we determined the cryo-EM structure of the full-length wild-type hP2X7R in the ATP-bound open state at 3.0 Å resolution (Fig. 2C–E and Supplementary Figs. 3 and 4 and Supplementary Table 1). As expected, the ATP-bound structure of the hP2X7R has a global architecture similar to the rP2X7R in the ATP-bound (RMSD = 0.8 Å at Cα carbons, PDB: 6U9W) and BzATP-bound (RMSD = 1.0 Å at Cα carbons, PDB: 8TRJ) open state conformations (Supplementary Fig. 8)[35,36]. The minimum pore radius of the hP2X7R in the ATP-bound open state is 2.5 Å, the same as the rP2X7R in the ATP-bound open state, and large enough to pass partially hydrated sodium ions such as those present in the closed pores of human, mouse, and rat P2X7Rs (Supplementary Fig. 9)[36,39]. Although there are structural similarities within the extracellular domains, the pore and cytoplasmic domains of the hP2X7R in the ATP-bound open state are rotated in comparison to the rP2X7R in the ATP-bound open state (Supplementary Fig. 8). Relative to the rP2X7R, TM1 of the hP2X7R

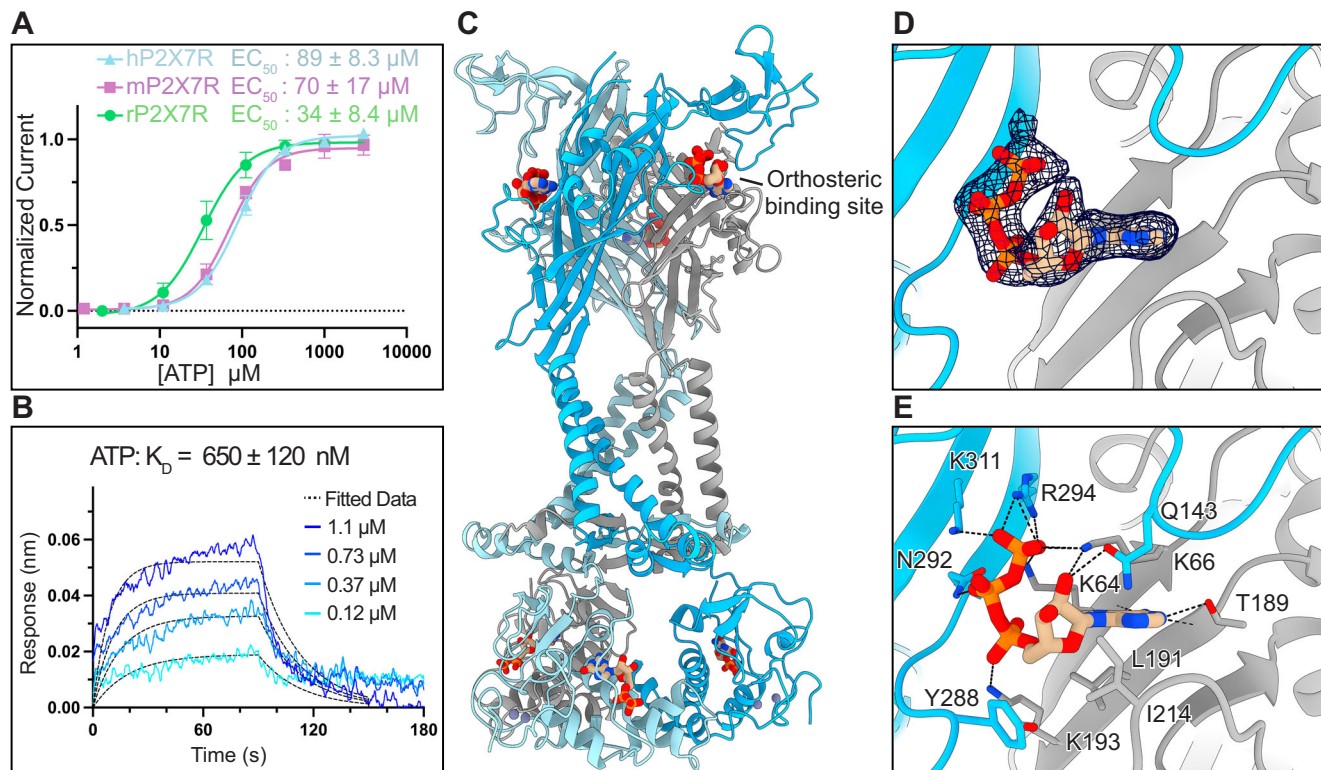

**Fig. 2 | ATP binds to the orthosteric pocket of the hP2X7R. A** Dose response curves from TEVC experiments measuring the activation of full-length wild-type human (blue), mouse (purple), and rat (green) P2X7Rs by ATP ($EC_{50} = 89 \pm 8.3\ \mu M$, $70 \pm 17\ \mu M$, and $34 \pm 8.4\ \mu M$, respectively). Data points and error bars represent the mean ± SD of normalized current, respectively, across triplicate experiments. **B** Representative BLI sensorgram for a dilution series of ATP (shades of blue) binding to biotinylated hP2X7R immobilized on streptavidin (SA) biosensors. Kinetic data were globally fit using a Langmuir 1:1 model to determine the equilibrium dissociation constant ($K_D$) of ATP to the hP2X7R as $K_D = 650 \pm 120$ nM, representing the mean ± SD across triplicate experiments. For kinetic analysis, a 90-s association time and a 60-s dissociation time were used for analysis. **C** Ribbon representation of the hP2X7R in the ATP-bound open state conformation at 3.0 Å colored by protomer (blue, light-blue, and gray), highlighting the orthosteric ATP-binding site. **D, E** Magnified view of the orthosteric ATP-binding site from **A**, highlighting the cryo-EM density for ATP (**D**) and residue interactions that coordinate ATP (**E**). ATP is coordinated by seven residues conserved across all P2XR subtypes (K64, K66, T189, K193, N292, R294, and K311) and four P2X7R subtype-specific residues (Q143, L191, I214, and Y288).

is rotated up in-plane by ~7° and TM2 is rotated up in-plane by ~6° with the hinges at the extracellular ends of TM1 and TM2, respectively (Supplementary Fig. 8C). These differences result in lateral displacements of 5.2 Å (distance between Cα carbons of residue 24) at the start of TM1 and 4.8 Å (distance between Cα carbons of residue 358) at the end of TM2 (Supplementary Fig. 8C). The rotation of TM2 is further propagated into a global rotation of the cytoplasmic ballast by ~10° in the hP2X7R relative to the rP2X7R (Supplementary Fig. 8C). This twist in the pore and cytoplasmic domain, much like the tightening of a spring, decreases the overall height of the hP2X7R by ~4 Å compared to the rP2X7R (distance between Cα carbons of residues 514 and 521), likely representing an inherent flexibility of the transmembrane and cytoplasmic domains in the ATP-bound open state conformation.

The orthosteric ATP-binding site of the hP2X7R is clearly visualized at 3.0 Å resolution (Fig. 2C, D). In the ATP-bound open state of the hP2X7R, ATP is coordinated by the seven residues that are conserved in the orthosteric pocket of all P2XR subtypes: K64, K66, T189, K193, N292, R294, and K311 (Fig. 2E and Supplementary Fig. 6). In addition, the hP2X7R-specific residues, L191, I214, and Y288, form hydrophobic interactions with the ribose group and the side chain of Q143 interacts with the 2′-hydroxy on the ribose group (3.3 Å) (Fig. 2E)[26]. Of these subtype-specific residues, I214 and Y288 in hP2X7Rs are not conserved across P2X7R orthologs (Supplementary Fig. 6). Residue Y288 in hP2X7Rs corresponds to a valine in mP2X7Rs and a phenylalanine in rP2X7Rs, causing differences that might affect the pharmacology of ATP binding (Supplementary Fig. 6). Residue I214 in hP2X7Rs and rP2X7Rs corresponds to a glycine in mP2X7Rs, introducing flexibility in the mouse ortholog that could explain some pharmacological differences (Supplementary Fig. 6)[30]. Indeed, it has previously been shown that R125, Q143, and I214 in the rP2X7R are the critical determinants of efficacy, potency, and full agonism for BzATP compared to ATP[35]. In our structure of the hP2X7R in the ATP-bound open state, residues R125, Q143, and I214 are in similar positions and rotameric conformations to the open state structure of the rP2X7R, including the flexible side chain of R125, which is solvent-facing and stubbed at the Cβ carbon[35]. The molecular details gleaned from the hP2X7R structure in the ATP-bound open state conformation will be crucial for structure-based drug design of high-affinity hP2X7R agonists.

## The classical allosteric pocket of the hP2X7R can fit larger cage alkyls

We sought to understand how an existing allosteric P2X7R antagonist binds to the human ortholog to determine its suitability as a starting candidate for structure-based drug design. We initially selected the adamantane-containing compound UB-ALT-P30 because of its simplicity, ease of synthesis, and potential for further functionalization[40,41]. UB-ALT-P30 consists of an adamantyl and a 2-chlorophenyl moiety connected by a hydrazide linker (Fig. 3A and Supplementary Fig. 1). In calcium influx assays, UB-ALT-P30 has a half-maximal inhibitory concentration ($IC_{50}$) of $17.8 \pm 8.3$ nM for the hP2X7R, $116 \pm 50$ nM for the rP2X7R, and $148 \pm 27$ nM for the mP2X7R, establishing its greater potency for the hP2X7R (Fig. 3B). Moreover, using a high concentration at which we can expect 100% inhibition of the hP2X7R (10 μM), UB-ALT-P30 is highly selective for the hP2X7R compared to the human P2X1

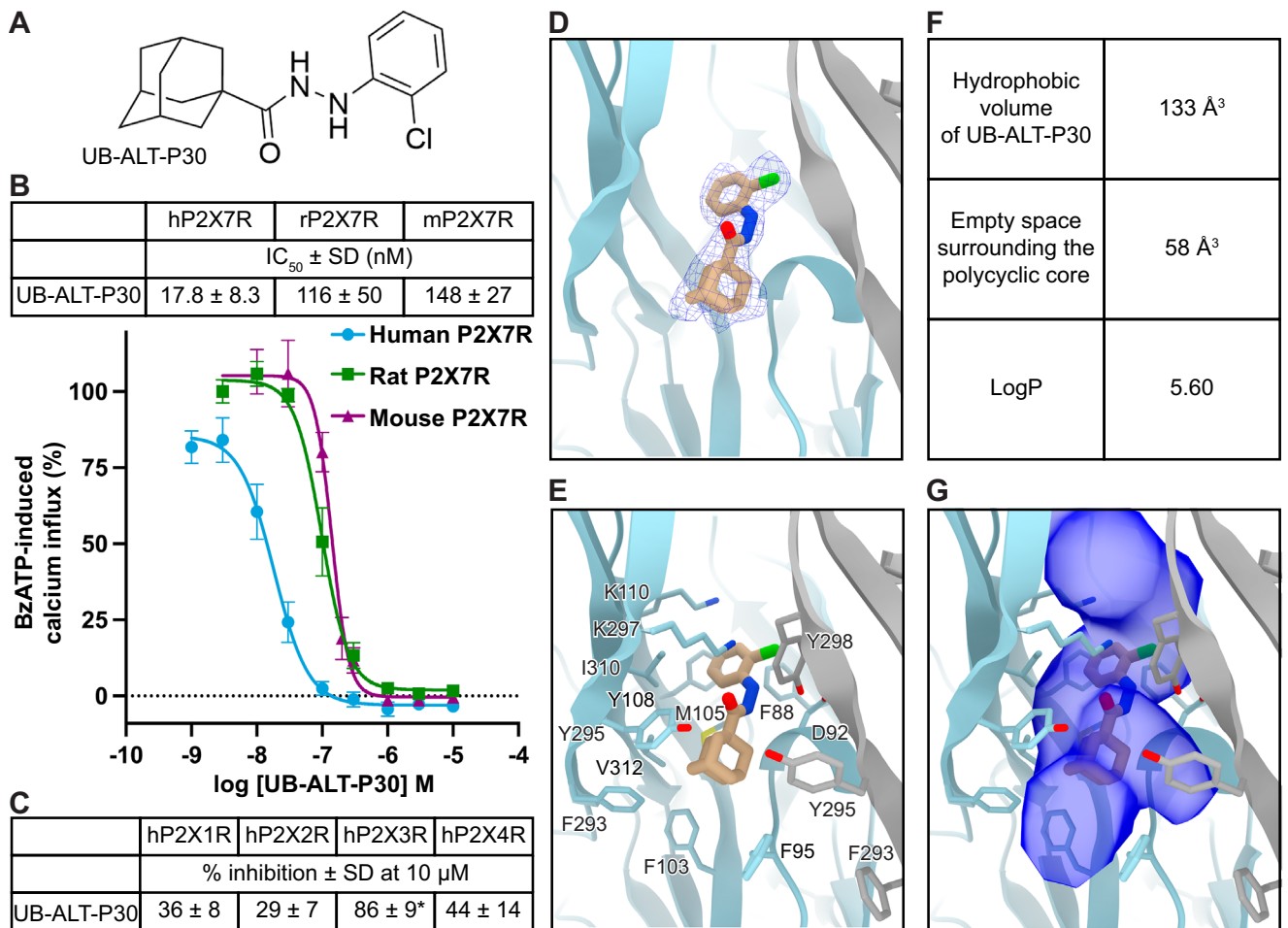

**Fig. 3 | Cryo-EM analysis and molecular dynamics of UB-ALT-P30 bound to the hP2X7R show that larger caged alkyls fit the classical allosteric pocket in the human ortholog. A** 2D chemical structure of UB-ALT-P30. **B** Concentration-dependent inhibition of agonist-induced calcium influx (IC$_{50}$ curves) for UB-ALT-P30 on human, rat, and mouse P2X7Rs. Receptors were recombinantly expressed in 1321N1 astrocytoma cells and activated by an EC$_{80}$ of the agonist BzATP. **C** Calcium influx assays were used to measure the potency of UB-ALT-P30 at hP2X1Rs, hP2X2Rs, hP2X3Rs, and hP2X4Rs stably expressed in 1321N1 astrocytoma cells. Receptors were activated by an EC$_{80}$ of ATP at the respective subtype (hP2X1R, 100 nM; hP2X2R, 1000 nM; hP2X3R, 100 nM; hP2X4R, 300 nM). *An IC$_{50}$ value of 707 ± 185 nM was estimated for UB-ALT-P30 at the hP2X3R. **B, C** Data represent mean ± SD of at least three independent experiments performed in duplicates. For each trace, the data were normalized to the calcium signal induced by the respective EC$_{80}$ concentration of agonist (when no antagonist was added). **D** Ribbon representation of the classical allosteric ligand-binding site in hP2X7R, located at the interface of two protomers (gray and light blue) with one molecule of UB-ALT-P30 (tan) shown with corresponding cryo-EM density (blue mesh) at 2.8 Å. **E** Residues in the classical allosteric ligand-binding site of the hP2X7R that interact with UB-ALT-P30. The polycyclic group of UB-ALT-P30 forms hydrophobic interactions with the receptor. **F** Computational values that describe the binding properties of UB-ALT-P30 within the classical allosteric pocket of the hP2X7R. The hydrophobic volume reported is the hydrophobic surface of the adamantyl group calculated via Maestro; Schrödinger. Unoccupied space surrounding the adamantyl was calculated via POVME3.0[42]. LogP values calculated via Maestro; Schrödinger. **G** Same view as **E** showing the accessible volume of the classical allosteric pocket in the hP2X7R as calculated by Fpocket using the UB-ALT-P30-bound inhibited state structure of the hP2X7R with the ligand removed[77]. Several voids flanking UB-ALT-P30, as well as vacant space above and below the ligand, could be targeted areas for further ligand development.

receptor (hP2X1R, 36 ± 8% inhibition), the human P2X2 receptor (hP2X2R, 29 ± 7%), and the human P2X4 receptor (hP2X4R, 44 ± 14%) (Fig. 3C). The ligand showed some inhibitory potency at the human P2X3 receptor (hP2X3R, estimated IC$_{50}$ of 707 ± 185 nM), which means that it is still 40-fold more potent at the hP2X7R versus the hP2X3R. The kinetics of UB-ALT-P30 binding to the hP2X7R were measured by TEVC, resulting in an on-rate of 0.0105 nM x min$^{-1}$ and an off-rate of 0.29 min$^{-1}$ (Supplementary Fig. 10). The high potency and moderate selectivity for the hP2X7R render UB-ALT-P30 a good initial compound for ligand development and optimization.

The high-resolution cryo-EM structure of UB-ALT-P30 bound to the hP2X7R (2.8 Å) reveals receptor-ligand interactions and provides key insights to optimize ligand design (Fig. 3D, E, Supplementary Fig. 3 and 4 and Supplementary Table 1). UB-ALT-P30 binds to the classical allosteric ligand-binding site, at the interface of upper body domains from neighboring protomers, in a shallow binding pose (Fig. 1A and Supplementary Fig. 2)[34]. The binding of ligands to the classical allosteric pocket is thought to prevent receptor movements necessary for transition to the ATP-bound open state[33,34]. UB-ALT-P30 binds to the hP2X7R in a similar pose as other adamantane-containing allosteric antagonists, generating a pruned RMSD of 0.6 Å compared to the structure of AZD9056 bound to the rP2X7R (PDB code: 8TR8) (Supplementary Fig. 11)[34]. However, important ortholog-specific interactions are also apparent.

The adamantyl moiety of UB-ALT-P30 is predominantly coordinated by hydrophobic interactions with the side chains of residues F95, F103, M105, F293, Y295, and V312 in the hP2X7R, which we confirmed with three replicas of 500 ns molecular dynamics (MD) simulations (Fig. 3E and Supplementary Fig. 12A, C). Of these residues, F95 is specific to the hP2X7R ortholog and located on a dynamic loop in the

upper body domain (residues 88–100) that is positioned differently in rat and human adamantane-based antagonist-bound structures (Fig. 3E and Supplementary Figs. 1, 2, 6, 11)[34]. In the structure of AZD9056 bound to the rP2X7R, the corresponding L95 is rotated towards the ligand, making extensive hydrophobic interactions[34]. In contrast, F95 in the hP2X7R points away from UB-ALT-P30, forming weak hydrophobic interactions and creating empty space below the molecule (Fig. 3E, F, G)[34]. Another ortholog-specific residue, V312, fills a hydrophobic cavity at one side of the adamantyl moiety in the hP2X7R (Fig. 3E and Supplementary Fig. 6). One additional hydrophobic cavity in the hP2X7R exists on the opposite side of the adamantyl moiety from V312, where three water molecules are coordinated by the backbone carbonyls of A296 (2.8 Å), R294 (3.2 Å), A91 (2.6 Å), and T94 (3.0 Å), as well as the backbone nitrogen of Y295 (3.3 Å). These cavities create ~58 Å$^3$ of empty hydrophobic space around the polycyclic core of UB-ALT-P30, presenting an opportunity to optimize this ligand by replacing the adamantyl with more precisely fitting scaffolds (Fig. 3F, G).

Additional receptor-ligand interactions in the hP2X7R can be observed closer to the extracellular surface of the classical allosteric pocket. The hydrazide linker of UB-ALT-P30 forms hydrogen bonds with the backbone carbonyl of D92 (2.8 Å) as well as the side chain hydroxyl of Y298 (3.4 Å) (Fig. 3E and Supplementary Fig. 12A, C). The chlorophenyl moiety is predominantly coordinated by hydrophobic interactions with the side chains of residues F88, M105, F108, and I310, as well as a cation-π interaction with the side chain ammonium of K297 (Fig. 3E and Supplementary Fig. 12A, C). Residues F108 in the hP2X7R and Y108 in the rP2X7R appear to play the same role, forming edge-to-face interactions with allosteric ligands[34]. Together, the empirical knowledge gained from identifying ligand-receptor interactions and empty space around the adamantyl moiety of UB-ALT-P30 in the classical allosteric pocket of the hP2X7R can be used to design ligand scaffolds with higher potency and greater selectivity (Fig. 3F, G)[42].

### UB-MBX-46 is a P2X7R antagonist with an improved scaffold

Given our detailed understanding of the molecular interactions between UB-ALT-P30 and the hP2X7R, we employed structure-based drug design to develop a more potent and selective antagonist. Notably, the adamantyl scaffold present in UB-ALT-P30 is also found in several clinically approved drugs and numerous preclinical and clinical candidates, including some P2X7R antagonists[26]. While the presence of other polycyclic scaffolds in approved drugs is rather limited, certain polycyclic hydrocarbons have outperformed adamantane for specific targets[43–46]. Therefore, we synthesized six compounds with alternative polycyclic cores of different sizes and shapes to replace the adamantyl scaffold of UB-ALT-P30 (Table 1). These six compounds were synthesized from the corresponding polycyclic carboxylic acids shown in Supplementary Fig. 13 following procedures reported for the synthesis of UB-ALT-P30[40,45,46]. All compounds were characterized by their spectroscopic data, melting point, exact mass, and elemental analysis or HPLC/UV (Supplementary Figs. 13–17 and Supplementary Methods).

The relative binding free energies (RBFEs) for each analog (relative to UB-ALT-P30) at the hP2X7R were determined in silico based on the corresponding perturbative transformation calculated using thermodynamic integration coupled with MD simulations (TI/MD) in phospholipid bilayers via Amber22 (Table 1 and Supplementary Fig. 18 and 19)[47–50]. Of all our compounds, only UB-MBX-46 and UB-MBX-P2 resulted in negative RBFEs ($\Delta\Delta G_{b,TI/MD}$), indicating that the perturbative transformations of these two compounds were energetically favorable (Table 1). The inhibitory potency of each compound was experimentally tested using ethidium bromide accumulation assays (Table 1) and generally correlated with the computational predictions (UB-MBX-46 being the most potent compound, followed by

UB-MBX-P2). Similarly, compounds with positive $\Delta\Delta G_{b, TI/MD}$ values were less potent, including UB-MBX-P1 and UB-ALT-P37 (Table 1). These data indicate that UB-MBX-46 is the most promising compound for further validation. The polycyclic tetracyclo [4.4.0.0$^{3,9}$.0$^{4,8}$] decane scaffold of UB-MBX-46 features two cyclopentane rings in a "frozen" envelope conformation and two unique cyclohexane rings in boat conformation. This scaffold is larger than adamantane and cubane (UB-ALT-P38), yet smaller than the pentacyclic moiety of UB-MBX-P1, and is unique since it has scarcely been used in medicinal chemistry (Table 1)[46].

### UB-MBX-46 is a potent and selective antagonist for the hP2X7R

To pharmacologically characterize the improved scaffold of UB-MBX-46, the compound was tested on different P2X7R orthologs and P2XR subtypes (Fig. 4A, Supplementary Fig. 1). In calcium influx assays, UB-MBX-46 has an inhibitory potency of 0.514 ± 0.035 nM for the hP2X7R and is ~35x more potent on the hP2X7R, ~3x more potent on the rP2X7R, and ~33x more potent on the mP2X7R than the starting compound UB-ALT-P30 (Fig. 4B). Furthermore, UB-MBX-46 is more potent at the hP2X7R than either the rP2X7R (~80-fold less potent) or the mP2X7R (~9-fold less potent) (Fig. 4B). The selectivity of UB-MBX-46 for other P2XR subtypes was also tested using calcium influx assays. Using a high concentration at which we can expect 100% inhibition of the hP2X7R (10 μM), UB-MBX-46 is much more selective for the hP2X7R compared to the hP2X1R (13 ± 10% inhibition), the hP2X2R (15 ± 9%), the hP2X3R (29 ± 16%), or the hP2X4R (41 ± 19%) (Fig. 4C). Thus, UB-MBX-46 is even more selective for the hP2X7R compared to other P2XR subtypes than UB-ALT-P30 (Figs. 3C and 4C). Finally, the kinetics of UB-MBX-46 binding to the hP2X7R were measured by TEVC, resulting in an on-rate of 0.0065 nM x min$^{-1}$ and an off-rate of 0.07 min$^{-1}$ (Fig. 4H, I). However, the virtually irreversible binding of UB-MBX-46 to the receptor within 10 min of wash-out does not correspond to the graphically determined $K_{off}$ for UB-MBX-46, so better approximations ($K_{off}$ of 0.001 min$^{-1}$) were obtained based on extrapolated IC$_{50}$ values (Supplementary Fig. 10).

We next investigated the molecular basis of the improved functional characteristics of UB-MBX-46 by determining the cryo-EM structure of the hP2X7R in the UB-MBX-46-bound inhibited state at 2.5 Å resolution (Fig. 4D, E; Supplementary Figs. 3 and 4 and Supplementary Table 1). Similar to the adamantane derivative, UB-MBX-46 binds to the classical allosteric ligand-binding site in a shallow binding pose (Fig. 1A and Supplementary Fig. 2)[34]. The two ligand-bound structures are very comparable, with a pruned RMSD of 0.5 Å however, the longer and larger scaffold of UB-MBX-46 relative to UB-ALT-P30 (hydrophobic volumes of 175 Å$^3$ vs 133 Å$^3$) binds deeper into the classical allosteric pocket and extends closer to the extracellular surface of the receptor (Figs. 3E, F and 4E, F). The hydrazide linker and chlorophenyl moieties of UB-MBX-46 are ~0.9 Å closer to the extracellular surface than UB-ALT-P30 (average measurements between equivalent chlorine atoms, nitrogen atoms on the hydrazide linker, and closest carbon atoms to the linker of the polycyclic moieties). Yet, due to the longer length of UB-MBX-46 and its larger polycyclic moiety, the ligand also extends deeper into the classical allosteric pocket by ~1.3 Å. As a result, there is only ~48 Å$^3$ of empty hydrophobic space surrounding the polycyclic core of UB-MBX-46; 10 Å$^3$ less empty space than around UB-ALT-P30 (Figs. 3F, G and 4F, G)[42].

The cryo-EM structures and MD simulations show that many of the receptor-ligand interactions associated with UB-ALT-P30 in the classical allosteric pocket of the hP2X7R are present with UB-MBX-46. Similar to the adamantyl moiety in UB-ALT-P30, the polycyclic core of UB-MBX-46 is predominantly coordinated by hydrophobic interactions with the side chains of residues F95, F103, M105, F293, Y295, and V312 (Fig. 4E and Supplementary Fig. 12B, D). However, due to the ligand's depth in the pocket, the larger polycyclic core of UB-MBX-46 is more able to form hydrophobic interactions with the human-specific

**Table 1 | Structure and characterization of P2X7R antagonists**

| Compound | Structure | IC$_{50}$ (nM) | $\Delta\Delta G_{exp}$[a] (kcal mol$^{-1}$) | $\Delta\Delta G_{b,TI/MD}$[b] (kcal mol$^{-1}$) | $\|\Delta\Delta G_{b,TI/MD} - \Delta\Delta G_{exp}\|$ (kcal mol$^{-1}$)[c] |
|---|---|---|---|---|---|
| UB-MBX-46 |  | 1.0 ± 0.1 | −2.37 ± 0.32 | −2.77 ± 0.06 | 0.40 ± 0.16 |
| UB-ALT-P30 |  | 47 ± 4 | 0 | 0 | 0 |
| UB-ALT-P36 |  | 477 ± 170 | 1.43 ± 1.93 | 1.33 ± 0.07 | 0.10 ± 0.97 |
| UB-MBX-P2 |  | 18 ± 2 | −0.59 ± 0.37 | −1.56 ± 0.05 | 0.97 ± 0.07 |
| UB-ALT-P37 |  | >1000 | >1.88 | 2.66 ± 0.06 | <0.78 |
| UB-MBX-P1 |  | 933 ± 156 | 1.84 ± 1.89 | 3.02 ± 0.07 | 1.18 ± 1.79 |
| UB-ALT-P38 |  | 430 ± 64 | 1.36 ± 1.50 | 4.32 ± 0.00 | 2.96 ± 1.13 |

IC$_{50}$ for each compound on the hP2X7R was measured by ethidium bromide accumulation assays. Each value represents the mean ± SD of normalized responses across triplicate experiments. IC$_{50}$ values of UB-MBX-P1 and UB-MBX-P2 have been reported previously[45]. RBFE values calculated with TI/MD method ($\Delta\Delta G_{b,TI/MD}$) for each perturbative transformation of UB-ALT-P30 to the analogs indicated in Table 1 represent mean ± SD for the three symmetry-related ligands bound to the receptor; each TI/MD calculation was performed in duplicate. Positive values indicate less favorable energetics, and negative values indicate favorable energetics.

[a]Experimental relative binding free energies ($\Delta\Delta G_{b,exp}$) computed using the experimental pIC$_{50}$ values and equation (1), as described in Supplementary Methods.
[b]RBFEs were calculated in the NVT ensemble and correspond to Helmholtz free energies ($\Delta\Delta A$) that are approximately equal to Gibbs free energies ($\Delta\Delta G$), see also Supplementary material[78].
[c]mue = 1.05 kcal/mol (mean unsigned error computed the mean of $|\Delta\Delta G_{b,TI/MD} - \Delta\Delta G_{b,exp}|$ values). [d]The correlation between $\Delta\Delta G_{b,TI/MD}$ and $\Delta\Delta G_{b,exp}$ values was 0.91.

residue F95. The hydrazide linker forms hydrogen bonding interactions with the backbone carbonyl of D92 (2.7 Å) as well as the side chain hydroxyl of Y298 (3.4 Å and 3.1 Å), creating one more hydrogen bond that is absent in the interactions with UB-ALT-P30 (Figs. 3E and 4E and Supplementary Fig. 12). Finally, the chlorophenyl moiety is predominantly coordinated by hydrophobic interactions with the side chains of residues F88, M105, F108, W167, and I310 as well as a cation-π interaction with the side chain ammonium of K297 (Fig. 4E and Supplementary Fig. 12B, D). UB-MBX-46 appears to better occupy the classical allosteric pocket in the hP2X7R due to the improved ligand-receptor interactions compared to UB-ALT-P30. Indeed, mutating the ortholog specific residues in the mP2X7R (residues 108 and 312) and the rP2X7R (residues 95, 108, and 312) influences the potency of UB-MBX-46 (Supplementary Fig. 20). Further, from TI/MD calculations, UB-MBX-46 also has a lower desolvation penalty to

bind to the receptor, indicating a preference for the hydrophobic pocket of the receptor over the solvent phase (Fig. 4F). Thus, more favored hydrophobic interactions surrounding the polycyclic core, more favorable hydrogen bonding interactions, and a lower desolvation penalty contribute to the higher potency of UB-MBX-46.

## Discussion

The P2X7R is a promising therapeutic target for numerous pathological inflammatory diseases, but pharmacological differences between receptor orthologs coupled with a lack of structural information have hampered efforts to develop a selective allosteric antagonist against the hP2X7R using structure-based drug design. The structural and functional data that we present here provide insights into the molecular differences between full-length wild-type rat, mouse, and human P2X7Rs —the three most relevant orthologs for drug development—in apo

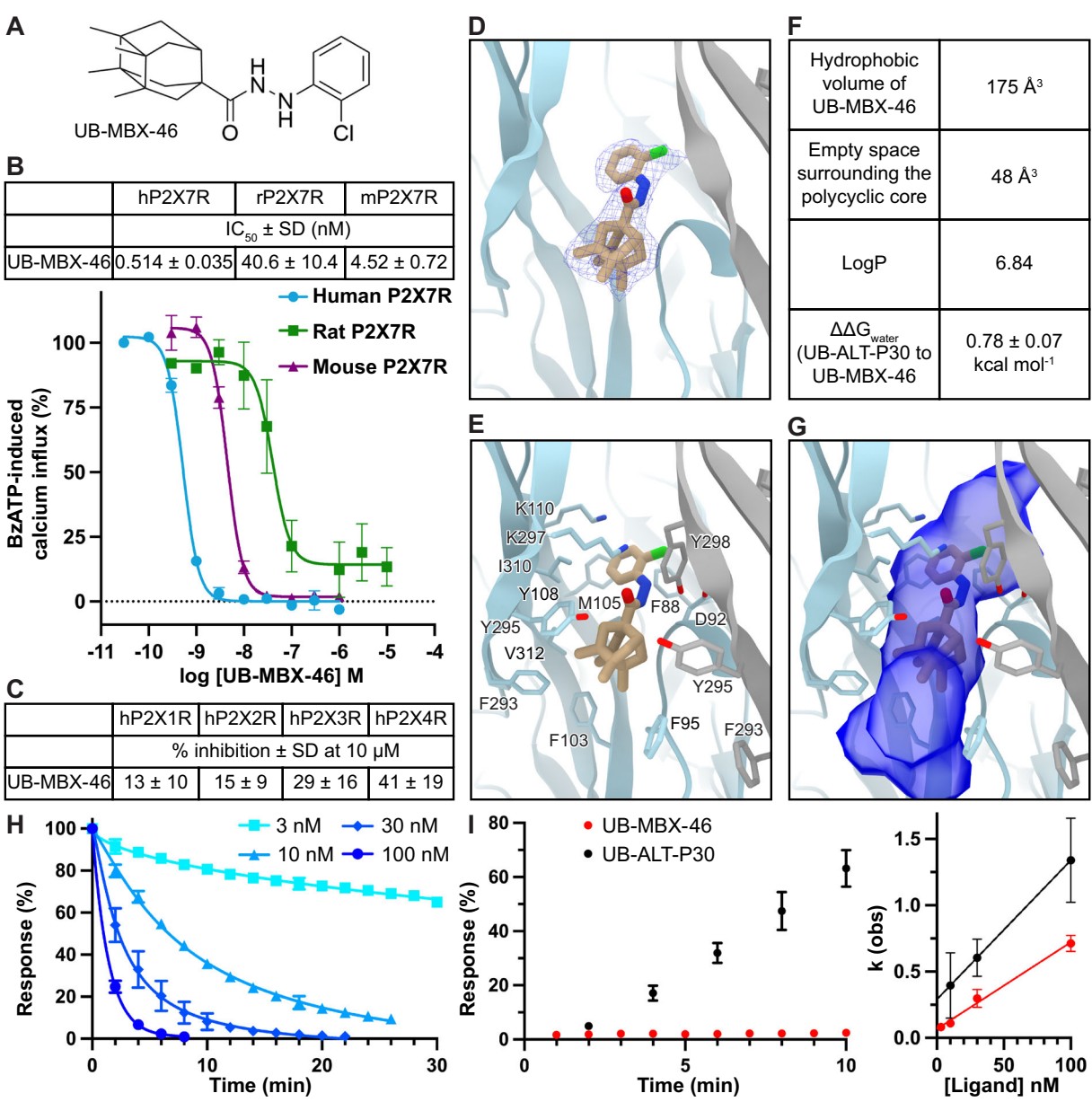

**Fig. 4 | UB-MBX-46 is a potent and selective allosteric antagonist for the hP2X7R. A** 2D chemical structure of UB-MBX-46. **B** Concentration-dependent inhibition of agonist-induced calcium influx (IC$_{50}$ curves) for UB-MBX-46 on human, rat, and mouse P2X7Rs. Receptors were recombinantly expressed in 1321N1 astrocytoma cells and activated by an EC$_{80}$ of the agonist BzATP. **C** Calcium influx assays were used to measure the potency of UB-MBX-46 at hP2X1Rs, hP2X2Rs, hP2X3Rs, and hP2X4Rs stably expressed in 1321N1 astrocytoma cells. Receptors were activated by an EC$_{80}$ of ATP at the respective subtype (hP2X1R, 100 nM; hP2X2R, 1000 nM; hP2X3R, 100 nM; hP2X4R, 300 nM). **B, C** Data represent mean ± SD of at least three independent experiments performed in duplicates. For each trace, the data were normalized to the calcium signal induced by the respective EC$_{80}$ concentration of agonist (when no antagonist was added). **D** Ribbon representation of the classical allosteric ligand-binding site in hP2X7R, located at the interface of two protomers (gray and light blue) with one molecule of UB-MBX-46 (tan) shown with its corresponding cryo-EM density (blue mesh) at 2.5 Å. **E** Residues in the classical allosteric ligand-binding site of the hP2X7R that interact with UB-MBX-46. The polycyclic group of UB-MBX-46 forms hydrophobic interactions with the receptor and the backbone carbonyl from D92, as well as the side chain hydroxyl of Y298 form hydrogen bonding interactions with the hydrazide linker. **F** Computational values that describe the binding properties of UB-MBX-46 within the classical allosteric pocket of the hP2X7R. The hydrophobic volume reported is the hydrophobic surface of the polycyclic group calculated via

Maestro, Schrödinger. Unoccupied space surrounding the polycyclic core was calculated via POVME3.0[42]. LogP values calculated via Maestro, Schrödinger. **G** Same view as **E** showing the accessible volume of the classical allosteric pocket in the hP2X7R calculated by Fpocket using the UB-MBX-46-bound inhibited state structure of the hP2X7R with the ligand removed[77]. UB-MBX-46 tightly fits the classical allosteric pocket of the hP2X7R, although unoccupied voids remain above and below the ligand. **H** Determination of binding kinetics by TEVC recordings. Normalized current responses to 5s-pulses of 300 μM ATP over time during continuous superfusion with the indicated concentrations of UB-MBX-46. All data are represented as mean ± SD with $n$ = 3, 4, 4, and 4 for 3 nM, 10 nM, 30 nM, and 100 nM respectively. **I** (*left*) Time course of antagonist dissociation. Normalized responses to 300 μM ATP after a 90–100% block by the antagonists are shown, highlighting virtually irreversible binding of UB-MBX-46 to the hP2X7R. (*right*) Experimentally determined on-rates ($k_{obs}$) were plotted against antagonist concentrations F to obtain an estimate for the off-rate constant $k_{off}$ (y-intercept) according to the formula $k_{obs} = k_{on}*F + k_{off}$. All data are represented as mean ± SD with n = 3 for UB-MBX-46 and $n$ = 5 for UB-ALT-P30 for the dissociation (*left*) and $n$ = 4 for UB-MBX-46 and $n$ = 7 for UB-ALT-P30 for the on-rates (*right*). Since there was minimal dissociation of UB-MBX-46 after 10 min, its graphically determined off-rate is clearly overestimated, and better values for $K_{off}$ were obtained based on the measured or extrapolated IC$_{50}$ values (Supplementary Fig. 10).

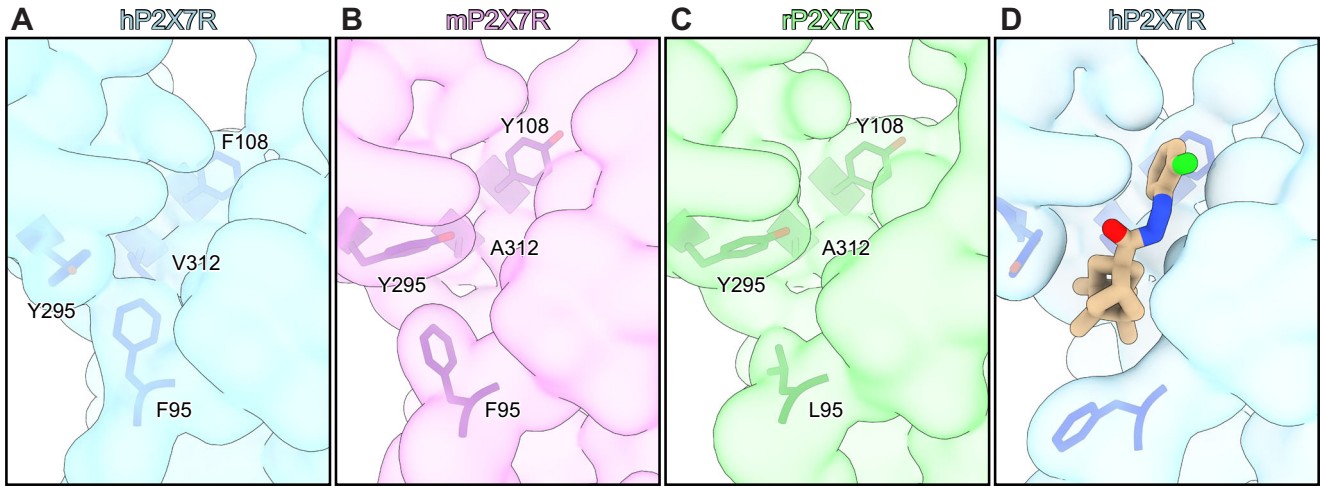

**Fig. 5 | A potent and selective scaffold for the allosteric modulation of the hP2X7R.** **A**–**D** Schematic representation of the unoccupied classical allosteric pocket in human, mouse, and rat P2X7R compared to the classical allosteric pocket for the hP2X7R occupied by UB-MBX-46. Surface rendering for one P2X7 protomer is shown to highlight the size and identity of ortholog-specific residues comprising the classical allosteric pocket. The classical allosteric pockets of the hP2X7R (**A**), the mP2X7R (**B**), and the rP2X7R (**C**) in the apo closed state conformation. The larger residues in the allosteric site of the hP2X7R appear to condense the pocket compared to the mP2X7R or the rP2X7R. Specifically, V312, F95, and the sterically rotated Y295 in the hP2X7R occupy more volume deep within the pocket. **D** The classical allosteric pocket of the hP2X7R is occupied by the potent and selective antagonist UB-MBX-46. The ligand is well tailored to the size and shape of the classical allosteric pocket in the human ortholog.

closed and ATP-bound open state conformations. The structure of the hP2X7R in complex with the known adamantane-based allosteric antagonist UB-ALT-P30 reveals ligand-receptor interactions within the classical allosteric pocket. We leveraged these data to optimize an antagonist scaffold tailored to the classical allosteric ligand-binding site in the human ortholog using structure-based drug design. After synthesizing and characterizing six antagonists with different polycyclic cores, we identified one (UB-MBX-46) that was highly potent and selective for the hP2X7R. The high-resolution cryo-EM structure confirmed that it binds to the classical allosteric binding site with optimized ligand-receptor interactions. With subnanomolar potency and high subtype selectivity, UB-MBX-46 is a promising compound with significant therapeutic potential for treating P2X7R-associated diseases.

CHS-binding sites were only evident in the apo closed state structure of the hP2X7R. No cryo-EM density for CHS was present in the structures of the mP2X7R and the rP2X7R in the apo closed state or the hP2X7R in the ATP-bound open state. The CHS-binding sites are located on the extracellular leaflet of the membrane, similar to the location of CHS in the apo closed state structure of the hP2X1R and likely represent cholesterol binding sites in vivo (Supplementary Fig. 21A, B)[51]. While characterizing the hP2X7R, we have observed that varying levels of CHS in the solubilization and purification process affect the stability of the cytoplasmic domain (Supplementary Figs. 2A and 21C, D). Insufficient levels of CHS throughout detergent solubilization and purification led to cryo-EM reconstructions with weak density for the cytoplasmic domains, in contrast to our experience with the mouse and rat orthologs (Supplementary Fig. 21C, D). Upon 3D-classification of the hP2X7R data, particles can be split into two distinct reconstructions with nearly identical extracellular and transmembrane domains but with the absence or presence of density for residues before TM1 or after TM2 (Supplementary Figs. 2A and 21C, D). These include residues that form the cytoplasmic cap, the C-cys anchor, and the cytoplasmic ballast (Supplementary Fig. 2A). We speculate that CHS molecules rigidify the detergent micelle to stabilize the palmitoyl groups on the C-cys anchor, which are known to permanently anchor the cytoplasmic cap[36]. Interestingly, the CHS molecules do not appear to interact with the palmitoyl groups within the model of the hP2X7R, suggesting other potential functional roles for cholesterol. Indeed, it is known that P2X7R activation and signaling are sensitive to cholesterol, serving as a negative

regulator of large-pore formation[52–55]. Thus, although we visualize the CHS-binding sites on the extracellular side of the lipid bilayer and have evidence that CHS stabilizes cytoplasmic domains on the intracellular side of the lipid bilayer, the functional effects of CHS on the hP2X7R need more investigation.

The identities of three residues and the rotameric conformation of a fourth residue within the classical allosteric pockets of human, mouse, and rat P2X7Rs confer ortholog-specific properties that explain why small molecules targeting P2X7Rs have significant interspecies variation. Among the three orthologs, residues F95, F108, and V312 in hP2X7Rs form the largest steric combination of amino acids. The larger side chain of V312 in the hP2X7R (A312 in the rP2X7R and the mP2X7R) forces an alternative rotameric conformation of Y295 compared to its conformation in the mP2X7R or the rP2X7R. As a result of these residue differences, the classical allosteric pocket of the hP2X7R is smaller than in the rP2X7R or the mP2X7R in the apo closed state conformation (Fig. 5A–C). In the presence of an antagonist, each of these residues plays an important role in coordinating ligand binding to either the rP2X7R or the hP2X7R (Supplementary Fig. 12C, D)[34]. Although no studies have previously investigated whether residues 95, 108, 295, and 312 affect ortholog-specific inhibition in functional assays, it is clear from the structural and functional data presented here that these residues influence ligand coordination in the classical allosteric pocket (Supplementary Figs. 7 and 20). The identities of residues 95, 108, 295, and 312 are also known to affect the inhibitory potency of other ligands and are therefore important considerations for the structure-based design of antagonists to target the hP2X7R[56,57].

To examine the stereoelectronic requirements for ligand-binding in the classical allosteric pocket of the hP2X7R, we determined the cryo-EM structure of an adamantane-based inhibitor, UB-ALT-P30, bound to the hP2X7R. In agreement with MD simulations, we observed a relatively poor fit of the molecule in the lipophilic cavity within the pocket and directed our efforts towards optimizing the polycyclic core of UB-ALT-P30, while retaining the hydrazide linker and chlorophenyl moiety, which appeared to have adequate interactions with the receptor. This motivation was triggered by recent research on polycyclic hydrocarbon structures in commercial drugs that occupy hydrophobic binding pockets and optimize drug potency[41,58]. Synthesis and testing of six different polycyclic moieties of different

sizes and shapes revealed a hydrophobic core that is either too big or too small resulting in poor inhibitory potency (Table 1). Initial efforts focused on adding methyl groups to the adamantane (compounds UB-ALT-P36 and UB-ALT-P37) to enhance hydrophobic interactions while also filling the binding pocket with minimal steric hindrance. However, this approach resulted in a significant reduction of antagonistic activity (Table 1). We also tested both smaller and larger alternative polycyclic structures. The larger of these permitted extensive surface interactions across the binding pocket, but the much less spherical shape introduced steric clashes that reduced its overall inhibitory activity. Another compound containing a bisnoradamantyl scaffold with a smaller polycyclic system of two methyl groups (compound UB-MBX-P2) provided efficient hydrophobic interactions and a two-fold increase in potency compared to UB-ALT-P30. Finally, the compound with the smallest polycyclic core (UB-ALT-P38) lacks stabilizing hydrophobic interactions needed for effective binding within the classical allosteric pocket, rendering it a poor inhibitor.

Our most successful compound, UB-MBX-46, has an intermediate-sized scaffold. For this compound, we modified a tetracyclo[4.4.0.0$^{3,9}$.0$^{4,8}$]decane core with four strategically placed methyl groups, generating a tetracyclic unit with a well-balanced size and hydrophobic profile (Fig. 5D). Consequently, UB-MBX-46 has sub-nanomolar potency ($IC_{50} = 510 \pm 40$ pM) with near irreversible binding and high selectivity for the P2X7R subtype (Fig. 4). The cryo-EM structure of the hP2X7R in complex with UB-MBX-46 and the associated MD simulations define the optimized ligand-receptor interactions that underlie its improved pharmacology (Fig. 5D). While unoccupied voids remain above and below the ligand suggesting further optimization is still possible, the flanking regions of UB-MBX-46 tightly fit the lateral constrictions within the classical allosteric pocket of the hP2X7R. Together, these results show why it is crucial to perform structure-based drug design on human orthologs to optimize pharmacokinetic properties. Moreover, as a result of this effort, we have provided a sub-nanomolar P2X7R antagonist with an improved ligand scaffold that can serve as a basis to develop even more potent and subtype–selective molecules with significant therapeutic potential.

## Methods

### Ethical statement

Unfertilized *Xenopus laevis* oocytes were purchased through Ecocyte Bioscience and kept at 18 °C until injection. This research complies with all relevant ethical regulations. All surgical procedures for isolation of *Xenopus laevis* oocytes were done in accordance with animal welfare laws, followed national and institutional guidelines for humane animal treatment, and complied with relevant legislation. Ecocyte Bioscience protocols help reduce the stress and harm on the laboratory animals, and appropriate aftercare, such as pain management, is employed to further minimize the impact of surgeries on the animals.

### Cell lines

SF9 cells were cultured in SF-900 III SFM (Fisher Scientific) at 27 °C. Cells of female origin were used for the expression of baculovirus.

HEK293 GNTI⁻ and TSA201 cells were cultured using Gibco Freestyle 293 Expression Medium (Fisher Scientific) at 37 °C supplemented with 2% v/v fetal bovine serum (FBS). HEK293 cells of female origin were used to express P2X7Rs.

Non-transfected human 1321N1 astrocytoma cells were purchased from Sigma-Aldrich (Taufkirchen, Germany). All stable cell lines were cultured at 37 °C and 5-10% $CO_2$ in Dulbecco's Modified Eagle Medium (Thermo Fisher Scientific, Dreieich, Germany) supplemented with 10% fetal calf serum, penicillin (100 U/mL), streptomycin (100 μg/mL), and the selection antibiotic geneticin G418 (800 μg/mL). All supplements were purchased from PAN Biotech (Aidenbach, Germany).

### Receptor constructs

The full-length wild-type hP2X7R and mP2X7R constructs used for protein expression contain a C-terminal HRV 3-C protease site, GFP, and a histidine affinity tag for purification. For structural determination, all tags were removed during purification, and no mutations or truncations were made to the receptor. For electrophysiology experiments to determine the apparent affinity of ATP, the hP2X7R-WT, mP2X7R-WT, and rP2X7R-WT constructs are unmodified, full-length wild-type receptors with no GFP, protease sites, or affinity tags present. For determination of binding kinetics, an N-terminally His-tagged hP2X7R (provided by G. Schmalzing, University of Aachen) in a modified pUC19 vector and a wild-type hP2X4R in a pNKS2 vector were used[59–61]. For calcium influx assays, a chimeric hP2X1R was employed, and for the hP2X3R, the S15V-mutant was used to prevent fast receptor desensitization[62,63].

### Receptor expression and purification

The full-length wild-type hP2X7R and mP2X7R constructs were expressed by baculovirus-mediated gene transfection (BacMam) using similar protocols as previously outlined for the rP2X7R[35,36]. Briefly, HEK293 GNTI⁻ or HEK293 cells were grown in suspension to a sufficient density and infected with P2 BacMam virus. Specifically, the ATP-bound and UB-ALT-P30-bound samples were prepared using HEK293 cells, while both apo and UB-MBX-46 samples were prepared using HEK293 GNTI⁻ cells. After overnight growth at 37 °C, sodium butyrate was added (final concentration of 10 mM) and the HEK293 GNTI⁻ cells shifted to 30 °C for an additional 48 hours. For the HEK293 cells, no sodium butyrate was added, and after overnight growth at 37 °C, the cells were shifted to 30 °C for an additional 48 hours. The cells were then harvested by centrifugation, washed with PBS buffer (137 mM NaCl, 2.7 mM KCl, 8 mM $Na_2HPO_4$, 2 mM $KH_2PO_4$), suspended in TBS (50 mM Tris pH 8.0, 150 mM NaCl) containing protease inhibitors (1 mM PMSF, 0.05 mg/mL aprotinin, 2 μg/mL Pepstatin A, 2 μg/mL leupeptin), and broken via sonication. Intact cells and cellular debris were removed by centrifugation, and then membranes were isolated by ultracentrifugation. Membranes were snap frozen and stored at −80 °C until use.

When ready, membranes were thawed, resuspended in TBS buffer containing 15% glycerol, dounce homogenized, and then solubilized. For the mP2X7R, the homogenized membranes were solubilized in 40 mM dodecyl-β-D-maltopyranoside (DDM or C12M) and 8 mM cholesterol hemisuccinate tris salt (CHS), while the hP2X7R was solubilized in 40 mM DDM and -17 mM CHS. After ultracentrifugation, the soluble fraction was incubated with TALON resin in the presence of 10 mM imidazole at 4 °C for 1–2 h. After packing into an XK-16 column, the purification column was washed with 2 column volumes of buffer (TBS plus 5% glycerol, 1 mM C12M, 0.2 mM CHS at pH 8.0) containing 20 mM imidazole, 10 column volumes containing 30 mM imidazole, and eluted with buffer containing 250 mM imidazole. Peak fractions containing the protein were concentrated and digested with HRV 3-C protease (1:25, w/w) at 4 °C overnight to remove GFP and the histidine affinity tag. To isolate the trimeric receptor from GFP, the digested protein was then ultracentrifuged and injected onto a Superose 6 10/300 GL column for size exclusion chromatography (SEC) that was equilibrated with 20 mM HEPES, pH 7.0, 100 mM NaCl, and 0.5 mM C12M. Fractions were analyzed by SDS-PAGE and fluorescence size exclusion chromatography (FSEC), pooled accordingly, and concentrated for cryo-EM grid preparation.

### Electron microscopy sample preparation

To prepare ligand-bound samples, the purified receptor was incubated with ligand at 3-4x the molar concentration of the monomer for antagonists or 900 μM ATP. After a one-hour incubation and shaking at 4 °C, cryo-EM grids were prepared for each sample. For all samples,

2.5 μL of solution was applied to glow-discharged (15 mA, 1 min) Quantifoil R1.2/1.3 300 mesh gold holey carbon grids and then blotted for 1.5 s under 100% humidity at 6 °C. The grids were flash frozen in liquid ethane using a FEI Vitrobot Mark IV and stored under liquid nitrogen until screening and large-scale data acquisition.

## Electron microscopy data acquisition
Cryo-EM data for all receptor orthologs and complexes were collected on Titan Krios microscopes (FEI) operated at 300 kV at the Pacific Northwest Center for Cryo-EM (PNCC). Datasets were acquired on Gatan K3 direct-electron detectors in super-resolution mode using defocus values that ranged between −0.8 and −1.5 μm. Movies were collected with between 44 and 50 frames and a total dose ranging between 42 and 45 e⁻/ Å². Datasets for both apo and antagonist-bound samples used an energy filter and nominal magnification of 130,000x, corresponding to a physical pixel size of ~0.648 Å/pixel. On the other hand, the ATP-bound dataset did not use an energy filter and was collected at a nominal magnification of 37,000x, corresponding to a physical pixel size of ~0.623 Å/pixel. Each dataset utilized "multi-shot" and "multi-hole" collection schemes driven by serialEM[64].

## Electron microscopy data processing
Cryo-EM movies were imported into cryoSPARC, and patch motion correction was completed, outputting all micrographs at the physical pixel size (~0.648 or 0.623 Å/pixel) (Supplementary Fig. 3 and Supplementary Table 1)[65]. Following estimation of the contrast transfer function (CTF) parameters, micrographs were curated and particles picked using 2D templates generated from a 3D volume. Particles were inspected, extracted, and sent directly to 3D classification, skipping 2D classification (Supplementary Fig. 3 and Supplementary Table 1). To remove bad particles, ab initio jobs generated reference volumes that were used for iterative heterogeneous classifications, ultimately yielding the final particle stacks. The final homogeneous particle stacks were used for non-uniform refinements with global and local CTF refinements to yield the consensus cryo-EM reconstructions which were B-factor sharpened in cryoSPARC (Supplementary Fig. 3, 4 and Supplementary Table 1)[66].

## Model building and structure determination
Homology models for apo closed state structures of the hP2X7R and the mP2X7R were generated from the apo closed state structure of the rP2X7R (PDB code: 8TR5), while the hP2X7R in the ATP-bound open state model was generated from the ATP-bound open state of the rP2X7R (PDB code: 6U9W) using SWISS-MODEL[35,36,67]. Each initial model was then docked into the corresponding cryo-EM map using ChimeraX[68,69]. Model building involved iterations of manual correction in COOT, followed by refinements in PHENIX[68,69]. All ligands were built and refined using eLBOW with protonation states corresponding to approximately pH 7[70]. Limited glycosylation, acylation, and palmitoylation were included in models when justified by the corresponding cryo-EM density. Model quality was evaluated by MolProbity (Supplementary Table 1)[71].

## Two-electrode voltage clamping (TEVC)
**Preparation of oocytes expressing P2X7Rs for EC₅₀ experiments.** Defolliculated *Xenopus laevis* oocytes were purchased from Ecocyte Bioscience and kept in Modified Barth's Solution (88 mM NaCl, 1 mM KCl, 0.82 mM MgSO₄, 0.33 mM Ca(NO₃)₂·4H₂O, 0.41 mM CaCl₂·2H₂O, 2.4 mM NaHCO₃, 5 mM HEPES) supplemented with amikacin 250 mg/L and gentamycin 150 mg/L. Oocytes were then injected with 50 nL of either full-length wild-type hP2X7R (40 ng/μL), mP2X7R (20 ng/μL), or rP2X7R (20 ng/μL) mRNA made from linearized pcDNA 3.1x according to the protocol provided in the mMessage mMachine kit (Invitrogen).

Injected oocytes were allowed to express for ~20 h before recording was performed.

**TEVC recordings for EC₅₀ experiments.** Data acquisition was performed using an Oocyte Clamp OC-725C amplifier and pClamp 8.2 software. Buffers were applied using a gravity-fed RSC-200 Rapid Solution Changer, flowing at ~5 mL/min. All experiments use Sutter filamented glass 10 cm in length with an inner diameter of 0.69 mm and an outer diameter of 1.2 mm to impale oocytes and clamp the holding voltage at −60 mV. Experiments were recorded in buffer containing 100 mM NaCl, 2.5 mM KCl, 0.1 mM EDTA, 0.1 mM flufenamic acid, and 5 mM HEPES at pH 7.4. All oocytes expressing P2X7R were facilitated with multiple applications of 100 μM ATP before any data was recorded.

**Dose response (EC₅₀) experiments.** Excitatory responses to a dilution series of ATP, ranging from 3 mM to ~1 μM, were performed to determine the EC₅₀ value of each P2X7R ortholog for ATP. Each evoked response was normalized to the signal evoked by the largest concentration of ATP, and the data were fitted in Prism 10 using the nonlinear regression named "EC₅₀, x is concentration" to afford EC₅₀ values. This value is then averaged amongst each singular condition and reported as the mean ± SD. ATP stocks were adjusted to a pH of 7.4 prior to performing TEVC experiments.

**Preparation of oocytes expressing hP2X7Rs for determination of antagonist binding kinetics.** *Xenopus laevis* frogs were from Nasco International (Fort Atkinson, WI) and kept at the Core Facility Animal Models (CAM) of the Biomedical Center (BMC) of LMU Munich, Germany (Az:4.3.2–5682/LMU/BMC/CAM) in accordance with the EU Animal Welfare Act. Ovaries were surgically extracted (ROB_54-011-AZ 2532.Vet_02-23-166), treated with collagenase (Nordmark, Uetersen, Germany 1.0–1.5 mg/mL, ≥2 at RT) and defolliculated (15-min treatment in Ca²⁺-free oocyte Ringer (90 mM NaCl, 1 mM KCl, 2 mM MgCl₂, 5 mM HEPES), respectively. 50 nL of cRNA (0.5 μg/μL) were injected using a Nanoject II injector (Science Products, Drummond) and oocytes were kept in ND96 (96 mM NaCl, 2 mM KCl, 1 mM MgCl₂, 1 mM CaCl₂, 5 mM HEPES, pH 7.4–7.5). The plasmid containing the hP2X7R was linearized with NotI-HF (NEB) and purified with the MinElute Reaction Cleanup Kit (Qiagen, Hilden, Germany). Capped cRNA was synthesized using the mMESSAGE mMACHINE™ T7 Transcription Kit (Thermo-Fisher Scientific, Schwerte, Germany) and dissolved in nuclease-free water (500 ng/μL). Injected oocytes were allowed to express for at least 36 hours.

**TEVC recordings for determination of antagonist binding kinetics.** Two-electrode voltage clamp recordings were performed at room temperature and a holding potential of −70 mV using a Turbo Tec-05X Amplifier (npi electronic, Tamm, Germany), a magnetic valve system, and CellWorks E 5.5.1 software. Currents were filtered at 1 kHz and digitized at 200 Hz. Electrode resistances were below 1.2 MΩ. A fast and reproducible solution exchange (<300 ms) was achieved with a 50-μl funnel-shaped oocyte chamber combined with a fast vertical solution flow fed through a custom-made manifold mounted immediately above the oocyte. Recordings were performed in ND96.

**Determination of antagonist binding kinetics.** A standard concentration of 300 μM ATP in low divalent recording buffer (ND96 with no MgCl₂ and 0.5 mM CaCl₂) was applied for 5 s in 2-min intervals. Between ATP applications, oocytes were continuously superfused with ND96. After obtaining reproducible agonist-evoked responses, solutions were switched to antagonist-containing recording solutions with and without ATP to determine association rates. Responses following antagonist application were normalized to the preceding, stabilized agonist responses. Dissociation was determined in 1- or 2 min intervals

when currents were reduced to <10%. Association curves were fit to the data by the equation % Response = (100-Plateau)*exp(-$k_{obs}$*Time) + Plateau, where Plateau indicates the response at equilibrium and $k_{obs}$ the experimentally determined on-rate. All data were analyzed using Prism software (GraphPad Software, Inc., Version 8.3.0, San Diego, CA) and are presented as means ± SD from at least three oocytes. Antagonists were dissolved in DMSO, and dilutions were made in ND96 (UB-ALT-P30) or DMSO (MBX-46). The final concentration of DMSO in the recording solution was ≤1%.

### Biolayer interferometry experiments

The hP2X7R was biotinylated, and BLI experiments were performed using similar protocols as previously outlined for measuring the affinity of ATP to the rP2X7R[35]. Briefly, BLI experiments were performed at 30 °C in 384-well tilted-bottom plates (Sartorius) with orbital shaking at 1000 r.p.m. on a ForteBio Octet RED384 instrument (ForteBio Data Acquisition software 11.0) using pre-equilibrated (0.22 μm filtered PBS at pH 7.4 and 0.5 mM DDM from Anatrace) SA biosensor tips (Sartorius) loaded with 50 μg/mL of biotinylated hP2X7R ligand or biocytin (Sigma-Aldrich) control for 1800 s[34,35]. The loaded SA sensors were blocked with 100 μg/mL of biocytin for 150 s and then washed with running buffer for 60 s. All loaded sensors were then baselined in running buffer for 120 s and dipped into wells containing threefold dilutions (10 to 0.12 μM for ATP) for 90 s and then returned to running buffer for the dissociation step for 300 s. After data collection, rate constants for association and dissociation were determined using ForteBio Data Analysis HT 11.0 evaluation software. To maximize signal between the large ligand and small analyte, the raw data were double reference subtracted[34,35]. For each interaction pair, the association rate constant ($k_a$) and the dissociation rate constant ($k_d$) were calculated with an average of three independent assays with at least four different concentrations that were globally fit to a 1:1 Langmuir binding model. The equilibrium dissociation constant ($K_D$) was calculated as the ratio of $k_d$ to $k_a$. The $K_D$, $k_a$, and $k_d$ values are reported as the average ± the standard deviation of three replicate datasets for each analyte, with each dataset using at least four independent traces. The analyzed data were exported and plotted in GraphPad Prism 9.0.

### Calcium influx assays

The inhibitory potencies of the antagonists UB-MBX-46 and UB-ALT-P30 were determined in calcium influx assays, as previously described[72,73]. For 1321N1 astrocytoma cells stably expressing the human, rat, or mouse P2X7Rs, or hP2X2Rs or hP2X4Rs, the fluorescent $Ca^{2+}$-chelating dye Fluo-4 acetoxymethyl ester (Fluo-4 AM, Thermo Fisher Scientific, Dreieich, Germany) was used. For 1321N1 astrocytoma cell lines recombinantly expressing the hP2X1Rs or hP2X3Rs, the FLIPR® Calcium 5 Assay Kit (Molecular Devices, San José, CA, USA) was employed. Human, rat, and mouse P2X7Rs as well as hP2X1Rs, hP2X2Rs, hP2X3Rs, and hP2X4Rs were recombinantly expressed in 1321N1 astrocytoma cells using a retroviral expression system[72–74].

On the first day, 45,000 cells per well (30,000 cells for the human P2X3 receptor-expressing cells) were seeded into a 96-well polystyrene microplate with a black, flat bottom in a final volume of 200 μL. After overnight incubation at 37 °C and 10% $CO_2$ (5% $CO_2$ for the human P2X3R-expressing cell line), the medium was removed by inverting the plate, and the cells were loaded with dye solution. Hanks' balanced salt solution (HBSS, Thermo Fisher Scientific, Dreieich, Germany) was used as the assay buffer for human P2X1R, P2X2R, P2X3R, and P2X4R. For the P2X7Rs, a different assay buffer was used: 150 mM sodium-glutamate, 5 mM KCl, 0.5 mM $CaCl_2$, 0.1 mM $MgCl_2$, 10 mM D-glucose, 25 mM 4-(2-hydroxyethyl)-1-piperazineethanesulfonic acid (HEPES), pH 7.4. The dye solutions were freshly prepared. The Fluo-4 AM solution contained 3 μM of Fluo-4 AM in assay buffer and 1% of the non-ionic detergent Pluronic F-127® (Sigma-Aldrich, Taufkirchen, Germany). The Calcium 5 dye, used for hP2X1Rs and hP2X3Rs, was prepared in HBSS buffer. Cells were loaded with Fluo-4 AM and incubated for 1 h at room temperature with gentle shaking (100 rpm), while those loaded with the Calcium 5 dye were incubated for 1 h at 37 °C. After incubation, the dye solution was carefully removed from the plate, and buffer was added to the cells for the evaluation of agonists. For determining $EC_{80}$ values for ATP (Roth, Karlsruhe, Germany) and BzATP (Jena Bioscience, Jena, Germany), respectively, dilution series were prepared in transparent 96-well plates (Boettger, Bodenmais, Germany) using the respective assay buffer. To determine $IC_{50}$ values of the antagonists, the dye solution was removed, and the dilution series of the antagonist dissolved in DMSO was added (final DMSO concentration: ≤1% for hP2X2Rs, hP2X4Rs, and hP2X7Rs, and ≤0.5% for hP2X1Rs and hP2X3Rs). The agonist solution was prepared in a 96-well transparent plate, obtaining the respective $EC_{80}$ values. As a control, assay buffer without the test compound was used. After 30 min of incubation with antagonist, the plates were measured using a fluorescence imaging plate reader NOVOstar (BMG Labtech GmbH, Offenburg, Germany) at an excitation wavelength of 485 nm and an emission wavelength of 520 nm, for 30 s at 0.4 s intervals. Calcium influx measurements were then initiated by automatic addition of 20 μL of agonist solution by the plate reader's pipetting device. Data were analyzed using Microsoft Excel and GraphPad Prism (Version 8.0, San Diego, CA, USA). $EC_{80}$ and $IC_{50}$ values were calculated using nonlinear regression with a sigmoidal dose-response equation.

To assess inhibitory potencies for mutant P2X7Rs, calcium influx assays in transfected HEK293T cells were performed following previously described literature, with slight modifications[72,73]. Briefly, HEK293T cells were trypsinized, adjusted to a density of 150,000 cells/mL, and transfected using Lipofectamine 2000 (ThermoFisher, Waltham, MA, USA) according to the manufacturer's protocol. About 0.5 μg of each cDNA construct (in pcDNA3.1x, per mL of cells) was mixed with 1.25 μL of Lipofectamine 2000 in OptiMEM (ThermoFisher, Waltham, MA, USA), incubated at room temperature for 20 min, and then added to the cells. The mixture was then seeded into black, flat-bottom 96-well polystyrene microplates (Corning CellBind, No. 3340, Kennebunk, Maine, USA) in a volume 200 μL/well and incubated for 48 h at 37 °C with 5% $CO_2$. Calcium assays were then performed as described above.

### Measurement of ethidium bromide accumulation in hP2X7R-expressing HEK293 cells

Human P2X7R-expressing HEK293 cells were cultured in a humidified atmosphere of 5% $CO_2$ at 37 °C in Dulbecco's Modification of Eagle's Medium (DMEM; Corning) supplemented with 10% (v/v) fetal bovine serum (Corning) and 1% (v/v) antibiotic–antimycotic (Gibco)[75]. To perform the assay, the DMEM was removed, and the HEK293 cells were washed with 3 mL of 1X DPBS (Dulbecco's Phosphate-Buffered Saline; Corning). After the removal of DPBS solution, the cells were detached from the dish with 1 mL of trypsin/EDTA (Gibco), and the cells were collected by centrifugation. The cells (density of $2.25 \times 10^7$) were resuspended in 4-(2-hydroxyethyl)-1-piperazineethanesulfonic acid (HEPES)-buffered salt solution consisting of 140 mM potassium chloride, 1 mM ethylene diamine tetraacetic acid (EDTA), 5 mM glucose, 20 mM HEPES, and 0.1 mM ethidium bromide (pH 7.4). The vehicles or appropriate range of concentrations of compounds (10 μL, pre-diluted in 10% (v/v) DMSO in DW from 10 mM stock) were added to each well of 96-well black plate (Corning), and 80 μL of the cell suspension was subsequently added to each well. BzATP (10 μL, pre-diluted in DW from 10 mM stock) was then added, and the final assay volume was maintained at 100 μL. After the incubation in 5% $CO_2$ at 37 °C for 2 h, the ethidium dye uptake was detected by measuring the fluorescence (excitation wavelength of 530 nm and emission wavelength of 590 nm) of each well using a CHAMELEON™ Multi-Technology Plate Reader. The antagonistic activities of compounds are expressed as the percentages relative to the maximum accumulation of ethidium bromide observed in the control group with the

stimulation of the hP2X7R by an agonist, BzATP. The $IC_{50}$ values of compounds as antagonists were calculated using nonlinear regression analysis using OriginPro 9.1 software.

## Synthesis of P2X7R antagonists

Compounds were synthesized as depicted in Supplementary Fig. 13. A series of new acyl hydrazides was synthesized starting from the corresponding carboxylic acids, following similar procedures to those previously reported for the synthesis of UB-ALT-P30[40,46]. The final products were purified by crystallization. Identity and purity were confirmed by $^{1}H$ and $^{13}C$ nuclear magnetic resonance (NMR), high-resolution mass spectrometry, elemental analysis, infrared, melting point, and HPLC/UV (Supplementary Figs. 13–17). See supplementary materials for detailed synthetic procedures, analytical data, NMR spectra, and, when appropriate, HPLC traces to demonstrate identity and purity for all products.

## Molecular dynamics simulations

The structures of full-length wild-type hP2X7R in the apo closed state and full-length wild-type hP2X7R in complex with UB-ALT-P30 or the UB-MBX-46 (with bound cholesterols) were utilized as starting models for 1 μs or 500 ns MD simulations at 310 K, respectively, with the Amber22 software[49]. Each protein structure, after suitable preparation, was inserted in a pre-equilibrated hydrated 1-palmitoyl-2-oleoyl-sn-glycero-3-phosphocholine (POPC) bilayer, expanding 30 Å from the furthermost vertex of the protein to the edge of the simulation orthorhombic box in all axes, with NaCl (0.150 M). The resulting lipid buffer contained ~479,000 atoms, consisting of 584 POPC lipids and ~123,500 water molecules. The dimensions of the simulation box were $150 \times 148 \times 226$ Å$^3$. Periodic boundary conditions were applied. We used the ff19sb to model the protein, the lipid21 force field to model the POPC lipids, GAFF2 to model the ligand, and the TIP3P model for waters and ions. After equilibration phase (see Supplementary Material), including restrained energy minimization, *NVT* MD simulation steps with Langevin thermostat (dynamics) using with a collision frequency of 2 ps$^{-1}$ and *NPT* MD simulation steps with Berendsen barostat using a coupling constant of 2 ps and the Langevin thermostat with a collision frequency of 2 ps$^{-1}$ were performed. Bonds involving hydrogen atoms were constrained by the SHAKE algorithm, and a time step of 2 fs with the leapfrog Verlet integrator was applied for unrestrained production runs at 310 K. Long-range electrostatics were calculated using Particle-mesh Ewald summation (PME), with a 1 Å grid, and short-range non-bonding interactions were truncated at 12 Å. A known MD simulation protocol was applied[76]. The MD simulations were performed in triplicate using random velocities in each replica.

For the TI/MD calculations performed with Amber22 software, minimized geometries of the complexes were then used for simulations at all $\lambda$ values[47,49]. Eleven $\lambda$ values were applied, equally spaced between 0.0 and 1.0.

Each MD simulation was heated to 310 K for 500 ps using the Langevin thermostat for temperature control. The Berendsen barostat was used to adjust the density. The 500 ps of NVT equilibration was followed by 2 ns NVT production simulation without restraints for 11 $\lambda$ windows to collect $\partial U/\partial \lambda$ data (see Supplementary Fig. 18). A 12-point Gaussian quadrature was used for the numerical integration of $\partial U/\partial \lambda$ to obtain all necessary $\Delta G$ values. Production simulations recalculated the potential energy at each $\lambda$ value every 1 ps for later analysis with MBAR. A known TI/MD simulation protocol was applied[76]. For each alchemical calculation of UB-ALT-P30 in complex with the full-length hP2X7R was applied dual topology, and the 1-step protocol was performed, which includes disappearing one ligand and appearing the other ligand simultaneously, and the electrostatic and van der Waals interactions are scaled simultaneously using softcore potentials from real atoms that are transformed into dummy atoms. The time step was 1 fs for all simulations, and SHAKE was not used. All TI simulations used the Berendsen thermostat with a coupling constant of 2 ps, except for the NPT equilibration step, which used the Langevin thermostat with a collision frequency of 2 ps$^{-1}$. The Berendsen barostat was used for NPT equilibration with a pressure relaxation time of 2 ps. Two repeats were performed for the 2 ns-TI/MD calculation for each alchemical transformation shown in Table 1. The alchemical calculation UB-ALT-P30 → UB-MBX-46 was also performed in the water and gas phase to account for the desolvation penalty. Particle Mesh Ewald Molecular Dynamics (pmemd) and energy minimization steps were performed using the Central Processing Unit (CPU) of workstations. The rest of the equilibration steps, including the unrestraint production, were run with AMBER22 software on RTX 4090 GPUs in lab workstations using pmemd.CUDA algorithm[49].

See supplementary material for detailed MD simulation protocols.

## Reporting summary

Further information on research design is available in the Nature Portfolio Reporting Summary linked to this article.

## Data availability

All cryo-EM density maps for the full-length wild-type mP2X7R in the apo closed state and the full-length wild-type hP2X7R in the apo closed, ATP-bound open, and antagonist-bound inhibited states have been deposited in the Electron Microscopy Data Bank (EMDB) under accession codes: EMD-47490 (apo closed hP2X7R), EMD-47491 (ATP-bound open hP2X7R), EMD-47492 (UB-ALT-P30-bound inhibited hP2X7R), EMD-47493 (UB-MBX-46-bound inhibited hP2X7R), and EMD-47494 (apo closed mP2X7R). The maps within these depositions include both half maps, sharpened/unsharpened maps, refinement masks, and any local refinements or locally sharpened maps that helped with model building. The corresponding coordinates for the structures have been deposited in Protein Data Bank under the PDB accession codes 9E3M (apo closed hP2X7R), 9E3N (ATP-bound open hP2X7R), 9E3O (UB-ALT-P30-bound inhibited hP2X7R), 9E3P (UB-MBX-46-bound inhibited hP2X7R), and 9E3Q (apo closed mP2X7R). All active compounds are available from the authors on reasonable request. First and last snapshots from MD simulations of complexes between full-length wild-type hP2X7R and UB-ALT-P30 or UB-MBX-46 and TI/MD simulations of the complexes shown in Table 1 can be found in the following GitHub repository: [https://github.com/georgioukyriakos/human-P2X7-receptor]. Additional PDB files were used for analysis including 8TR5 (apo closed rP2X7R), 6U9W (ATP-bound open rP2X7R), and 8TR8 (AZD9056-bound inhibited rP2X7R)[34–36]. The source data underlying Table 1, Figs. 2A, B, 3B, C and 4B, C, H, I, and Supplementary Figs. 8D, 10A, B, D and 20A, B, C are provided as a Source Data file. Source data are provided with this paper.

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

## Acknowledgements

We thank O. Davulcu, C. Yoshioka, and C. López at PNCC for access and microscopy assistance. Electron microscopy grid screening was performed at the Multiscale Microscopy Core within Oregon Health & Science University (OHSU). The authors would like to acknowledge the contributions of the OHSU Biophysics Shared Resource Core (Research Resource ID: RRID: SCR_022744) in facilitating this work. We thank L. Anson for comments on the manuscript. A portion of this research was supported by NIH grant U24GM129547 and performed at the PNCC at OHSU and accessed through EMSL (grid.436923.9), a DOE Office of Science User Facility sponsored by the Office of Biological and Environmental Research. This research was supported by the National Heart, Lung, and Blood Institute (R00HL138129, S.E.M.), the National Institute of General Medical Sciences (DP2GM149551, S.E.M.), and the American Heart Association (24PRE1195450, A.C.O.). Part of this work was funded by the Spanish *Ministerio de Ciencia, Innovación y Universidades*, MICIU/AEI/10.13039/501100011033 and by ERDF/EU: grant PID2023-147004OB-I00 (to S.V.). The Kolocouris Lab thanks Chiesi Hellas for funding (SERG No. 10354). C.E.M. is grateful to the German Federal Ministry of Education and Research (BMBF) for support (Biopharma Neuroallianz, 0315606B), the Deutsche Forschungsgemeinschaft (SFB 1328) for supporting this work, and the COST Action CA21130 "P2X receptors as a therapeutic opportunity (PRESTO)". We thank Andhika B. Mahardhika for support in some of the biological experiments. A.N. is supported by the Deutsche Forschungsgemeinschaft (DFG, German Research Foundation) Project-ID 335447717 - SFB 1328, project A15. The Mansoor Lab would like to thank Steve Janik and Sheryl Manning, Barbara Allen and Jim Batzer, and Randy and Barbara Lovre for their generous support.

## Author contributions

A.C.O., A.L.T., S.V., A.K., and S.E.M. designed the project. A.C.O. performed the cryo-EM sample preparation, data collection, data processing, and built the models. A.C.O. performed and analyzed the BLI experiments. A.C.O. performed and analyzed the TEVC electrophysiology dose-response experiments. J.N., F.G.W., and C.E.M. prepared stable cell lines as well as transiently transfected HEK293 cells expressing human, rat, and mouse P2X7Rs, performed the

calcium influx assays, and analyzed the data. G.K., S.L., and Y.K. performed and analyzed the ethidium bromide accumulation experiments. A.L.T., M.B.X., and S.V. designed, synthesized, and purified the P2X7R antagonists. E.T., K.G., and A.K. performed and analyzed the MD simulations as well as additional calculations; E.T. and K.G. contributed equally. J.S. and A.N. performed and analyzed the TEVC experiments to determine on- and off-rates. A.C.O., A.L.T, A.N., J.N., C.E.M., A.K., S.V., and S.E.M. wrote and edited the manuscript. All authors approved the manuscript.

## Competing interests

The authors declare no competing interests.
