## [Transparent Peer Review file · Nature Communications]

A unique polycyclic scaffold identified by structure-based drug design effectively inhibits the human P2X7 receptor

Corresponding Author: Dr Steven Mansoor

Version 0:

Reviewer comments:

Reviewer #1

(Remarks to the Author)

This manuscript presents several important findings that significantly advance our understanding of the human P2X7 receptor. First, it reports the first high-resolution structures of the human P2X7 receptor in both the apo state and in complex with ATP and two allosteric inhibitors, UB-ALT-P30 and UB-MBX-46. It also reports the first high-resolution structure of the mouse P2X7 receptor in the apo state. Second, it delineates key molecular differences between the rat, mouse, and human P2X7 receptor structures—three critical orthologs in drug development—highlighting the necessity of utilizing human structures for structure-based drug design. Third, it reports the development of a novel human P2X7 allosteric ligand with subnanomolar inhibitory potency, featuring a polycyclic tetracyclodecane moiety, an uncommon scaffold in medicinal chemistry. This new ligand has the potential to facilitate P2X7-targeted drug development, particularly considering that several previous candidates, despite progressing to clinical trials, have failed to reach the market.

Fourth, the study resolves two cholesterol hemisuccinate (CHS) binding sites per protomer in the apo human P2X7 structure. These sites are located at the apex of the interface between the TM1 and TM2 helices. Notably, the CHS molecules are not resolved in the ATP-bound state nor are they present in the mouse or rat P2X7 apo structures. The authors provide compelling evidence that insufficient CHS levels during solubilization and purification resulted in cryo-EM maps where both the TM and intracellular domains were not resolved, suggesting that CHS molecules play a stabilizing role in these domains in the apo state. The absence of resolved CHS molecules in mouse and rat P2X7 is puzzling, as several studies have previously suggested that cholesterol from rat and mouse P2X7 may act as an allosteric inhibitor of P2X7. One possible explanation is that cholesterol binds loosely to rat and mouse P2X7 while exhibiting a stronger binding affinity to human P2X7.

Overall, the authors present a strong body of evidence by integrating multiple state-of-the-art techniques, including cryo-EM structural analysis, two-electrode voltage-clamp electrophysiology, bio-layer interferometry, functional calcium and dye-uptake assays, organic synthesis, and molecular dynamics simulations. The manuscript is well-written, logically structured, and supported by high-quality figures.

I do not have major concerns, as this work has been convincingly executed. However, I have a few minor points that could improve the manuscript:

1. Throughout the manuscript, the authors claim that UB-MBX-46 binds to human P2X7 with picomolar potency. However, based on their functional assays, the IC₅₀ values range between 1 nM (dye-uptake) and 0.5 nM (calcium influx), far from 1 pM. I suggest stating instead that UB-MBX-46 exhibits subnanomolar potency.

2. Line 234: The authors report the percentage inhibition at 10 μM UB-ALT-P30 for other P2X7 paralogues: hP2X1 (36%), hP2X2 (29%), hP2X3 (86%), and hP2X4 (44%), and compare these to hP2X7, where complete inhibition (100%) is expected at the same concentration. They then claim that the "strong selectivity and high potency for hP2X7 render UB-ALT-P30 a good starting compound for ligand development and optimization." However, the term "strong" seems inappropriate, given the substantial inhibition observed for hP2X3 (~90%) and the lack of full inhibition curves for these receptors. I recommend modifying this wording or providing additional data to substantiate the claim of strong selectivity.

3. Figure 2B: The fitting curves for the bio-layer interferometry (BLI) sensorgram data appear suboptimal, particularly in the

dissociation phase of ATP. Could the authors discuss this point, given that the BLI methods are not described in the Methods section? Additionally, they should comment on the KD values obtained via BLI and the EC50 values derived from TEVC recordings, particularly in conditions where hP2X7 currents were facilitated by multiple ATP applications.

4. In the MD methods, the term "f-hP2X7" is used. Could the authors clarify its significance relative to hP2X7? Does "f-hP2X7" correspond to the apo hP2X7 structure with the cytoplasmic ballast domain removed?

5. Line 393: The authors cite two references (52 and 53) investigating the effect of cholesterol on human P2X7. However, while reference 53 examined the impact of cholesterol on human macrophages, it did not directly assess its effect on the human P2X7 receptor itself but rather on the mouse P2X7 receptor. Therefore, I suggest that the authors include additional references discussing the effect of cholesterol on rat or mouse P2X7. For example, the following references could be added: PMID: 34207150 and PMID: 37446075.

Reviewer #2

(Remarks to the Author)

This manuscript represents an important study in the P2X7 receptor field. Firstly, the authors obtain structures of both human and mouse P2X7 receptor orthologs and thereby provide critical insights into the pronounced species-dependent pharmacology that has long puzzled the field. Second, they use a computationally aided medicinal chemistry campaign to identify and characterize UB-MBX-46 as a potent and selective antagonist of the human P2X7 receptor. The data are solid and the conclusions largely justified. While there are no major issues with the work, a number of clarifications, controls and additional discussion should be included before the manuscript is considered for publication.

- It is well established that alterations of the human P2X7 receptor intracellular domain have functional consequences. The authors should include TEVC experiments to compare WT P2X7 and the construct used in Cryo-EM (containing "C-terminal GFP, a 3C protease site, and a histidine affinity tag for purification").
- Related to the above points: the authors report a relatively low EC50 for ATP (<100 micromolar) in their TEVC experiments. Although this is not without precedent, many other groups have reported EC50 values closer to 1 mM ATP (see PMID 25695577 and others). Please comment on the potential origins of these not insignificant differences.
- Line 234: the authors state "strong selectivity" with regards to the effect on hP2X3. But in light of the numbers (86 ± 9% inhibition for P2X3 at a concentration eliciting 100% inhibition on P2X7), the wording should be revised
- Line 291: the authors state "The relative binding free energies (RBFES) for each UB-ALT-P30 analog on the hP2X7R were determined in silico based on the corresponding perturbative transformation calculated using thermodynamic integration coupled with MD simulations (TI/MD) in phospholipid bilayers viaAmber22" However, it is not clear which structure is used here. Is it hP2X7R – UB-ALT-P30 bound?
- Line 315: UB-MBX-46 is still inhibiting hP2X3 around 30% and hP2X4 around 40% at concentrations eliciting 100% inhibition in P2X7. If it binds when the receptor is in the open state, given that hP2X7 needs higher concentration of ATP to open, the compound will likely affect hP2X3 and hP2X4 before hP2X7 under physiological conditions. Please comment.
- Line 321: it is not clear what the conformation of hP2X7 with UB-MBX-46 structure is – closed state?
- Line 424ff: it is slightly ambiguous if the data from the compounds these methyl additions have been described in the results section? Please clarify or add, if applicable
- Line 584: has the pH of ATP solutions been adjusted prior to experiments?
- Has UB-MBX-46 been tested in electrophysiology? It would be relevant to compare the results from the ethidium-bromide uptake with a method with more kinetic resolution, i.e. patch-clamp or TEVC. Same goes for the selectivity between subtypes, i.e. at least verify by testing against one of the other subtypes
- Do the authors classify UB-MBX-46 as an inhibitor/antagonist or a negative allosteric modulator (NAM)? The nomenclature is confusing throughout the paper and should be clarified.
- Figure 1: include example traces for TEVC recordings (here or elsewhere)
- Fig 3 and 4: include raw data for calcium influx assay and consider leaving out panels F, or better explain the relevance. Only the DDG number in Fig 5F is useful without further context. The other (absolute) numbers are not very informative
- Figure 5: please render the surface a bit more transparent and include in-figure labeling regarding the ortholog to orient the reader (similar to Fig 1)
- Have the authors tried to obtain the structure with ATP + UB-MBX-46? This is important because given the calcium and ethidium bromide uptake (where the compounds are applied after the application of agonists), the small molecule binders could be NAMs, so the relevance of a structure with only the NAM is unclear
- It would be of high relevance to show and compare the pore profiles of the structures, i.e. UB-MBX-46 bound, UB-ALT-P30 bound, ATP-bound and APO etc
- Suppl data: please include yields for the chemistry
- Suppl data, page 30 line 930: the rationale of using closed state models to define more precisely what happens with a NAM?

Reviewer #3

(Remarks to the Author)

Authors use a combination of Cryo-EM, MD simulation, calcium influx assay and TEVC to obtain structure-function relationships of P2X7Rs across multiple orthologous and at apo-closed and ATP-bound "open" states, shedding light on the species-specific differences that have confounded previous drug discovery efforts. The authors also use a structure-based design approach to obtain a pM affinity binder at the human ortholog and have convincingly rationalized its selectivity across

species.

The cryo-EM structures are of high quality and are properly presented. The associated functional assays and RBFE calculations have been well performed and also support the authors underlying claims. This study will undoubtedly be very useful to the scientific community and I commend the authors for an overall excellent piece of work; however, I have some concerns with part of the MD simulation work before I can recommend publication. I also have some more minor suggestions related to punctuation, sentence structure and general clarity of the presented data.

Major comments:

- Line 250- 251: A single run of an MD simulation does not confirm the presence of these interactions and is merely anecdotal. As the outcome of any MD simulation is highly dependent on the initial conditions, multiple replicas must be performed to obtain statistics/ confidence about a binding mode and I ask the authors to please include this additional data.
- The authors claim that they have solved an ATP-bound open state conformation, however it is not obvious that this structure is indeed an open-state. MD simulations that demonstrate ion flux through the pore would allow you to confidently annotate your structure as 'open' and these simulations should also be performed. Additionally, some measurement of pore diameter should also be included.
- Lines 294-296: These results are interesting and certainly agree with experimental data, however some convergence metrics of the FEP calculations must be included in the extended/supplementary data.
- The authors seem to have performed their RBFE calculations in the NVT ensemble. This means that they have actually obtained the Helmholtz free energy, rather than the Gibbs free energy. This distinction is unlikely to have a big impact on the results and indeed, their data agrees with experiment regardless. That being said, $\Delta\Delta G$ is therefore an incorrect label, and $\Delta\Delta A$ should be used instead.

Minor comments:

- In the abstract, lines 40-41, "Interspecies variation among existing antagonists...". This sentence is worded poorly and should be rephrased. Perhaps something like: "Understanding the species-specific pharmacological effects of existing antagonists has been challenging, largely due to differences in receptor orthologs and the limited molecular data available for comparison"
- Introduction, lines 88-90 refer to classical and extended allosteric binding sites, however Panel A at Extended Data Fig 2 has not been labeled with the location of these sites.
- Results, lines 160-162: The claim here that V312 forces Y295 to adopt a different conformation is not obvious from your figure. I can see that there's a difference in Y295 at human vs rat and mouse orthologs, but V312 appears to be far away from Y295 and on the basis of the figure alone, doesn't support this particular claim. Could you elaborate on this, perhaps with a clearer figure. This claim is also stated again in the figure legend.
- Lines 243-246. An overlay of these structures would be nice to see in the extended data with ortholog specific differences highlighted.
- Lines 282-283: "although no other singular polycyclic hydrocarbon has the success rate of adamantane in medicinal chemistry", success rate of what exactly? Could you please clarify.
- Extended data figure 8 legend: No need to include "embedded in POPC bilayers", you've already written that in the methods and it's superfluous here.
- Line 666: What is the "f" in f-hP2XR. I couldn't see where in the manuscript that was defined.
- Line 675: "ions" are specified but not their species or specific concentration.
- Lines 675: How long was your equilibration phase for?
- Line 678: "a time step of 1fs and time of 2fs" doesn't make sense. What was the time step of your simulation?
- Berendsen barostat has been specified for NPT, but what thermostat was used? Was it also Langevin?
- Line 689: "A known MD simulation protocol was applied". Please explicitly describe the methods used in the supplementary.
- Line 703: I can't see this in the supplementary.
- Supplementary figure 6 is not referenced in the text.

• Lines 761-762 (figure 1): The GDP and Zn²⁺ ions are not shown in Extended data figure 1 as referenced here. Extended Data Fig. 1 shows the 2D chemical structures of P2X7R antagonists.

• It is unclear to me why BzATP-induced calcium influx was normalized to an EC80, and to just one of the species? Why is hP2X7R normalized to this value in figure 3? Are these response curves properly normalized in figure 4?

Version 1:

Reviewer comments:

Reviewer #1

(Remarks to the Author)

The authors have satisfactorily addressed all the points I raised. I have no additional comments.

Reviewer #2

(Remarks to the Author)

The authors have an excellent job addressing the concerns. No further comments.

Reviewer #3

(Remarks to the Author)

I am satisfied with the authors updates and recommend this manuscript for publication. Congratulations again to the authors on this thoroughly enjoyable read and important contribution!

Reviewer #4

(Remarks to the Author)

The authors collected over 8000 Images, processed them in cryoSPARC, and have good resolutions to fit most of the structure well. The processing is standard. Overall, the structures are well determined.

Question:

Is the magnification number for ATP hP2X7R correct? 37 seems wrong.

Suggestions:

I would have liked to see Supplementary Figure 7 with densities to show that the changes in conformation observed are well determined in the map. The same is true for Supplementary Figure 21—for cholesterol.

The general comment is that the allosteric binding site is very similar to that observed in P2X3. While, the focus of the paper is on P2X7 - these are very similar in overall structures which is what leads to all the cross reactivities and therapeutic difficulties.

REVIEWER COMMENTS

Reviewer #1 (Remarks to the Author):

This manuscript presents several important findings that significantly advance our understanding of the human P2X7 receptor. First, it reports the first high-resolution structures of the human P2X7 receptor in both the apo state and in complex with ATP and two allosteric inhibitors, UB-ALT-P30 and UB-MBX-46. It also reports the first high-resolution structure of the mouse P2X7 receptor in the apo state. Second, it delineates key molecular differences between the rat, mouse, and human P2X7 receptor structures—three critical orthologs in drug development—highlighting the necessity of utilizing human structures for structure-based drug design. Third, it reports the development of a novel human P2X7 allosteric ligand with subnanomolar inhibitory potency, featuring a polycyclic tetracyclodecane moiety, an uncommon scaffold in medicinal chemistry. This new ligand has the potential to facilitate P2X7-targeted drug development, particularly considering that several previous candidates, despite progressing to clinical trials, have failed to reach the market.

Fourth, the study resolves two cholesterol hemisuccinate (CHS) binding sites per protomer in the apo human P2X7 structure. These sites are located at the apex of the interface between the TM1 and TM2 helices. Notably, the CHS molecules are not resolved in the ATP-bound state nor are they present in the mouse or rat P2X7 apo structures. The authors provide compelling evidence that insufficient CHS levels during solubilization and purification resulted in cryo-EM maps where both the TM and intracellular domains were not resolved, suggesting that CHS molecules play a stabilizing role in these domains in the apo state. The absence of resolved CHS molecules in mouse and rat P2X7 is puzzling, as several studies have previously suggested that cholesterol from rat and mouse P2X7 may act as an allosteric inhibitor of P2X7. One possible explanation is that cholesterol binds loosely to rat and mouse P2X7 while exhibiting a stronger binding affinity to human P2X7.

Overall, the authors present a strong body of evidence by integrating multiple state-of-the-art techniques, including cryo-EM structural analysis, two-electrode voltage-clamp electrophysiology, bio-layer interferometry, functional calcium and dye-uptake assays, organic synthesis, and molecular dynamics simulations. The manuscript is well-written, logically structured, and supported by high-quality figures.

I do not have major concerns, as this work has been convincingly executed. However, I have a few minor points that could improve the manuscript:

1. Throughout the manuscript, the authors claim that UB-MBX-46 binds to human P2X7 with picomolar potency. However, based on their functional assays, the IC₅₀ values range between 1 nM (dye-uptake) and 0.5 nM (calcium influx), far from 1 pM. I suggest stating instead that UB-MBX-46 exhibits subnanomolar potency.

- Thank you for pointing this out. As suggested, we have now replaced all mention of “picomolar” to “subnanomolar”.

2. Line 234: The authors report the percentage inhibition at 10 μ M UB-ALT-P30 for other P2X7 paralogues: hP2X1 (36%), hP2X2 (29%), hP2X3 (86%), and hP2X4 (44%), and compare these to hP2X7, where complete inhibition (100%) is expected at the same concentration. They then claim that the “strong selectivity and high potency for hP2X7 render UB-ALT-P30 a good

starting compound for ligand development and optimization." However, the term "strong" seems inappropriate, given the substantial inhibition observed for hP2X3 (~90%) and the lack of full inhibition curves for these receptors. I recommend modifying this wording or providing additional data to substantiate the claim of strong selectivity.

- Thank you for this suggestion. We have modified the word "strong" to now say "moderate" selectivity, which we agree is more accurate. We also now state the IC_{50} of UB-ALT-P30 at the hP2X3R (707 ± 185 nM) in the text (line 237) to convey the ligand is 40-fold less potent at the hP2X3R than the hP2X7R.

3. Figure 2B: The fitting curves for the bio-layer interferometry (BLI) sensorgram data appear suboptimal, particularly in the dissociation phase of ATP. Could the authors discuss this point, given that the BLI methods are not described in the Methods section? Additionally, they should comment on the KD values obtained via BLI and the EC50 values derived from TEVC recordings, particularly in conditions where hP2X7 currents were facilitated by multiple ATP applications.

- We thank the reviewer for catching this omission! We have now added a detailed description of the BLI methods to the Methods section of the manuscript (lines 653-673).

We agree that the BLI fits are not ideal. However, we want to stress that we are right at the edge of detectability for this method (ATP is a 554 Da small molecule binding to a greater than 200 kDa receptor), and we are unaware of any other study that directly measures the binding of ATP to the human P2X7R. The fitting software in the Octet machine used for BLI measurements assumes a 1:1 model. Unfortunately, the software does not take into account any cooperativity between the three orthosteric pockets. While the fits are not perfect, the results are in agreement with expected values. If the reviewer would like, we can add some disclaimers to this effect.

4. In the MD methods, the term "f-hP2X7" is used. Could the authors clarify its significance relative to hP2X7? Does "f-hP2X7" correspond to the apo hP2X7 structure with the cytoplasmic ballast domain removed?

- We apologize for the confusion here. The term "f-hP2X7" is an internal nomenclature to reflect the fact that the full protein was used including the ballast domain for the MD simulations. We have now corrected this to say "full-length wild-type hP2X7R" in the revised manuscript, both in the Main Text and in the Supplementary Information.

5. Line 393: The authors cite two references (52 and 53) investigating the effect of cholesterol on human P2X7. However, while reference 53 examined the impact of cholesterol on human macrophages, it did not directly assess its effect on the human P2X7 receptor itself but rather on the mouse P2X7 receptor. Therefore, I suggest that the authors include additional references discussing the effect of cholesterol on rat or mouse P2X7. For example, the following references could be added: PMID: 34207150 and PMID: 37446075.

- We thank the reviewer for the excellent suggestions and apologize for omitting these obviously relevant studies. We have now cited both references.

Reviewer #2 (Remarks to the Author):

This manuscript represents an important study in the P2X7 receptor field. Firstly, the authors obtain structures of both human and mouse P2X7 receptor orthologs and thereby provide critical insights into the pronounced species-dependent pharmacology that has long puzzled the field. Second, they use a computationally aided medicinal chemistry campaign to identify and characterize UB-MBX-46 as a potent and selective antagonist of the human P2X7 receptor. The data are solid and the conclusions largely justified. While there are no major issues with the work, a number of clarifications, controls and additional discussion should be included before the manuscript is considered for publication.

- It is well established that alterations of the human P2X7 receptor intracellular domain have functional consequences. The authors should include TEVC experiments to compare WT P2X7 and the construct used in Cryo-EM (containing “C-terminal GFP, a 3C protease site, and a histidine affinity tag for purification”).

- We thank the reviewer for these comments, and we completely agree. There is a very nice study that demonstrated how both the N- and C-terminal domains of the human P2X7R modulate ATP sensitivity (Klapperstuck et al., 2001; PMC2278689). This is why we go out of our way to remove the GFP and the purification tags before any of the structural biology work is performed. Thus, the wild-type TEVC and cryo-EM constructs are the same. We have now clarified the language in the methods to make this important point clearer (Lines 496-501).

- Related to the above points: the authors report a relatively low EC₅₀ for ATP (<100 micromolar) in their TEVC experiments. Although this is not without precedent, many other groups have reported EC₅₀ values closer to 1 mM ATP (see PMID 25695577 and others). Please comment on the potential origins of these not insignificant differences.

- Thank you for these comments. It is well established that the presence of divalent cations impacts the EC₅₀ values for P2XRs. We chose to perform these EC₅₀ experiments with EDTA present (in the absence of divalent cations) and observe values that are consistent with the literature under these conditions. We have added text to lines 171-172 to make this clearer.

- Line 234: the authors state “strong selectivity” with regards to the effect on hP2X3. But in light of the numbers (86 ± 9% inhibition for P2X3 at a concentration eliciting 100% inhibition on P2X7), the wording should be revised

- Thank you for this suggestion. We have modified the word “strong” to now say “moderate” selectivity, which we agree is more accurate. We also now state the IC₅₀ of UB-ALT-P30 at the hP2X3R (707 ± 185 nM) in the text (line 237) to convey the ligand is 40-fold less potent at the hP2X3R than the hP2X7R.

- Line 291: the authors state “The relative binding free energies (RBFES) for each UB-ALT-P30 analog on the hP2X7R were determined in silico based on the corresponding perturbative transformation calculated using thermodynamic integration coupled with MD simulations (TI/MD) in phospholipid bilayers viaAmber22” However, it is not clear which structure is used here. Is it hP2X7R – UB-ALT-P30 bound?

- The Reviewer is right here. We apologize for the confusion. The RBFs for each UB-ALT-P30 analog were calculated relative to UB-ALT-P30 on the hP2X7R. We clarified this in the main text in the revised manuscript (Line 298).

- Line 315: UB-MBX-46 is still inhibiting hP2X3 around 30% and hP2X4 around 40% at concentrations eliciting 100% inhibition in P2X7. If it binds when the receptor is in the open state, given that hP2X7 needs higher concentration of ATP to open, the compound will likely affect hP2X3 and hP2X4 before hP2X7 under physiological conditions. Please comment.

- Thank you for this interesting comment. We speculate these allosteric antagonists do not bind to the ATP-bound open state of the P2X7R. The proposed mechanism for classical allosteric P2X7R antagonists is a "peg-in-a-hole" model. Thus, the antagonist binds in the apo closed state conformation and prevents ATP from inducing the conformational changes required for pore opening.

- Line 321: it is not clear what the conformation of hP2X7 with UB-MBX-46 structure is – closed state?

- Thank you for pointing this out. We have changed the sentence to make it clear that the structure is of the hP2X7R in the UB-MBX-46-bound inhibited state (Line 335). The pore in the inhibited state is closed with the same architecture as the apo closed state. We have also added MOLE plots (pore radius throughout the receptor structure) as Supplementary Fig. 9 to show the state of the receptor in each conformational state (apo closed, antagonist-bound inhibited, and ATP-bound open). We think this should hopefully make things much clearer.

- Line 424: it is slightly ambiguous if the data from the compounds these methyl additions have been described in the results section? Please clarify or add, if applicable

- Thank you for identifying this point of confusion. We have now added a notation to indicate that we are specifically referring to compounds UB-ALT-P36 and UB-ALT-P37 (Line 442). In addition, we have added a callout to Table 1 which highlights their poor inhibitory potencies.

- Line 584: has the pH of ATP solutions been adjusted prior to experiments?

- Thank you. We have clarified this concern by adding to the methods: "ATP stocks were adjusted to a pH of 7.4 prior to performing TEVC experiments." (Lines 611-612).

- Has UB-MBX-46 been tested in electrophysiology? It would be relevant to compare the results from the ethidium-bromide uptake with a method with more kinetic resolution, i.e. patch-clamp or TEVC. Same goes for the selectivity between subtypes, i.e. at least verify by testing against one of the other subtypes

Thank you for these very astute comments. We performed TEVC experiments to determine the on- and off-rates for both UB-ALT-P30 and UB-MBX-46 (now shown in Fig. 4 and Supplementary Fig. 10). There was essentially no dissociation of UB-MBX-46 after 10 minutes (please see Fig. 4I). In addition, we tested the ability of UB-MBX-46 to antagonize hP2X4R using TEVC and it was negligible (please see Supplementary Fig. 10B). So, we now have assayed the properties of UB-MBX-46 by electrophysiology experiments, ethidium bromide uptake assays, and calcium influx assays.

- Do the authors classify UB-MBX-46 as an inhibitor/antagonist or a negative allosteric modulator (NAM)? The nomenclature is confusing throughout the paper and should be clarified.

- We thank the Reviewer for these questions. Based on the structures and the functional experiments, we can definitively say that UB-MBX-46 is an allosteric antagonist, as it binds to the classical allosteric pocket (a site distinct from the orthosteric ATP-binding site) and inhibits subsequent ATP-induced activation. The structure of the UB-MBX-46-bound inhibited state is virtually identical to the apo closed state structure (as opposed to the binding of UB-MBX-46 inducing an alternative structural conformation). Thus, we are refraining from using the term negative allosteric modulator. We have now added a clarifying sentence early in the manuscript to indicate when we refer to an “antagonist”, we mean “allosteric antagonist” (Line 64).

- Figure 1: include example traces for TEVC recordings (here or elsewhere)

- Thank you for this suggestion. We have now added a representative TEVC trace for each ortholog in Supplementary Fig. 8 (Supplementary Fig. 8D).

- Fig 3 and 4: include raw data for calcium influx assay and consider leaving out panels F, or better explain the relevance. Only the DDG number in Fig 5F is useful without further context. The other (absolute) numbers are not very informative

We thank the Reviewer for the suggestions. All of our raw data for TEVC, calcium imaging, and ethidium bromide uptake assays will be included in the source data file upon acceptance.

As mentioned in the manuscript, the pocket where UB-ALT-P30 binds is predominantly influenced by hydrophobic interactions, which are supported by a high hydrophobic volume of the compounds and Log P values. Additionally, calculations confirmed the presence of empty space around the polycyclic ring, indicating the need for larger compounds with increased hydrophobic volume and Log P to better fill this space. The three parameters that we show in the figure—hydrophobic volume, empty space surrounding the polycyclic core, and Log P—are crucial for predicting which compound will fit better into the binding pocket, ultimately improving the potency of UB-ALT-P30. Our approach was to increase the hydrophobic volume of the nonpolar molecule from 133 Å in UB-ALT-P30 to 175 Å in UB-MBX-46 (line 340). By doing so, we ensured better space occupancy, reducing the empty space around the polycyclic core and enhancing binding and potency (lines 345-348). Additionally, higher Log P values indicate increased hydrophobicity, which further supports these improvements. We hope this helps clarify and provides context for why we are including this information in the figures.

- Figure 5: please render the surface a bit more transparent and include in-figure labeling regarding the ortholog to orient the reader (similar to Fig 1)

- Thank you for the suggestion. We have now added the ortholog labeling for clarity. We have tried multiple different transparency levels, and we feel the current one looked the best. If we make the surface more transparent, we lose definition of the sidechain density. We would like to keep it as it is, if possible.

- Have the authors tried to obtain the structure with ATP + UB-MBX-46? This is important because given the calcium and ethidium bromide uptake (where the compounds are applied after the application of agonists), the small molecule binders could be NAMs, so the relevance of a structure with only the NAM is unclear

- Thank you for this suggestion. One point of clarification: for these studies, the agonists are applied AFTER the antagonists (please see line 703-712 and lines 737-741). We have now made this clearer by updating line 708.

Regarding trying to obtain a structure with ATP+UB-MBX-46, we have not tried this yet. It is a bit complicated, though, because it is not clear if we should add the ATP first and then add the antagonist. Or, add the antagonist first and then add the ATP. We currently have limited time on the microscope and feel this experiment is presently outside the scope of the current manuscript. But it is something we would like to try in the future.

- It would be of high relevance to show and compare the pore profiles of the structures, i.e. UB-MBX-46 bound, UB-ALT-P30 bound, ATP-bound and APO etc

- Thank you for this excellent suggestion. We agree with you and have added Supplementary Fig. 9 to highlight the pore differences and RMSD's between the various structures.

- Suppl data: please include yields for the chemistry

- Thank you for the suggestions. The yield values are in the Supplementary data. Specifically, they can be found after the detailed procedure for each synthesis, for example, lines 1041, 1064, 1085, and 1105.

- Suppl data, page 30 line 930: the rationale of using closed state models to define more precisely what happens with a NAM?

- The cryo-EM structure of hP2X7R in the apo closed state was used as starting models for MD simulations because allosteric antagonists do not directly compete with agonists at the orthosteric binding site but rather influence receptor dynamics by stabilizing fewer active conformations. The closed state represents a conformation where the receptor is in its inactive or less active form. The only structural difference in the receptor between the apo closed state conformation and the antagonist-bound inhibited-state conformation is the movement of a loop (residues 88-100) that occurs to allow ligand binding.

Reviewer #3 (Remarks to the Author):

Authors use a combination of Cryo-EM, MD simulation, calcium influx assay and TEVC to obtain structure-function relationships of P2X7Rs across multiple orthologous and at apo-closed and ATP-bound “open” states, shedding light on the species-specific differences that have confounded previous drug discovery efforts. The authors also use a structure-based design approach to obtain a pM affinity binder at the human ortholog and have convincingly rationalized its selectivity across species.

The cryo-EM structures are of high quality and are properly presented. The associated functional assays and RBE calculations have been well performed and also support the authors underlying claims. This study will undoubtedly be very useful to the scientific community and I commend the authors for an overall excellent piece of work; however, I have some concerns with part of the MD simulation work before I can recommend publication. I also have some more minor suggestions related to punctuation, sentence structure and general clarity of the presented data.

Major comments:

- Line 250- 251: A single run of an MD simulation does not confirm the presence of these interactions and is merely anecdotal. As the outcome of any MD simulation is highly dependent on the initial conditions, multiple replicas must be performed to obtain statistics/ confidence about a binding mode and I ask the authors to please include this additional data.

- Thank you for the comment. In the original version of the manuscript, we transferred a lot of the MD simulation methods to the supplemental materials and thus, it was likely unclear in the main text. We performed all the MD and TI/MD simulations in duplicate using random velocities for each replica. Further, we treated the three allosteric ligand-binding sites within the trimeric receptor as independent systems, so there are three internal replicates for each simulation.

We clarify this now in the revised manuscript in the main text (line 256) and in the Methods Section (line 780). We have added Supp. Fig. 12C,D that shows the statistics for selected MD simulations.

- The authors claim that they have solved an ATP-bound open state conformation, however it is not obvious that this structure is indeed an open-state. MD simulations that demonstrate ion flux through the pore would allow you to confidently annotate your structure as ‘open’ and these simulations should also be performed. Additionally, some measurement of pore diameter should also be included.

- Thank you for these suggestions. We have now added Supplementary Fig. 9 to highlight the pore differences, pore diameters, and RMSD’s between the various structures. This figure shows that indeed, the ATP-bound structure has an open pore; the pore is large enough to pass partially hydrated sodium ions from the extracellular fenestrations into the cell through the cytoplasmic fenestrations. Previous MD simulations have been performed on the hP2X3R to demonstrate ion and water flux through the pore (Mansoor et al., 2016, Nature). The open state

structures of human P2X7R and human P2X3R are so similar that we do not see the utility in repeating these simulations.

- Lines 294-296: These results are interesting and certainly agree with experimental data, however some convergence metrics of the FEP calculations must be included in the extended/supplementary data.

- We thank the reviewer for this comment. We now include the convergence plots of $\Delta V/\Delta t$ in the revised manuscript as Supplementary Fig. 18. In addition, we have added discussion about it in the main text (lines 299-302) and in the Supplementary File (lines 1286-1297).

- The authors seem to have performed their RBFEn calculations in the NVT ensemble. This means that they have actually obtained the Helmholtz free energy, rather than the Gibbs free energy. This distinction is unlikely to have a big impact on the results and indeed, their data agrees with experiment regardless. That being said, $\Delta\Delta G$ is therefore an incorrect label, and $\Delta\Delta A$ should be used instead.

This comment offers a great point of discussion. The Reviewer is correct that the calculation affords the Helmholtz free energy $\Delta\Delta A_b$ values and not the Gibbs free energy $\Delta\Delta G_b$ values. However, the Gibbs free energy is related to the Helmholtz free energy according to the equation: $G=A+PV$. Assuming the volume change upon binding to be negligible, which is often the case at 1 atm due to the incompressibility of the system, then the Gibbs free energy ΔG_{bind} is approximately equal to the Helmholtz free energy ΔA_{bind} (see relevant discussion in the reference below). The deviation of $\Delta\Delta A_{bind}$ from $\Delta\Delta G_{bind}$ is further reduced since we calculated differences in ΔA_{bind} values. Therefore, $\Delta\Delta A_{bind}$ should be considered almost indistinguishable from $\Delta\Delta G_{bind}$. We add a footnote comment in Table 1 and add the above-mentioned reference and this discussion in the Supplementary Materials in the Methods Section for the TI/MD simulations. Based on the above explanations, we would prefer to keep the $\Delta\Delta G_{b,TI/MD}$ label for the calculated values for straightforward comparison with $\Delta\Delta G_{b,exp}$ values.

Reference: Mey, A. S. J. S., Allen, B. K., Bruce McDonald, H. E., Chodera, J. D., Hahn, D. F., Kuhn, M., Michel, J., Mobley, D. L. ., Naden, L. N., Prasad, S., Rizzi, A., Scheen, J., Shirts, M. R., Tresadern, G., & Xu, H. (2020). Best Practices for Alchemical Free Energy Calculations [Article v1.0]. Living Journal of Computational Molecular Science, 2(1), 18378).

Minor comments:

- In the abstract, lines 40-41, "Interspecies variation among existing antagonists...". This sentence is worded poorly and should be rephrased. Perhaps something like: "Understanding the species-specific pharmacological effects of existing antagonists has been challenging, largely due to differences in receptor orthologs and the limited molecular data available for comparison"

- Thank you. We have added your suggested sentence, which sounds much better.

- Introduction, lines 88-90 refer to classical and extended allosteric binding sites, however Panel A at Extended Data Fig 2 has not been labeled with the location of these sites.

- Thank you for catching this. We have removed the callout to Supplementary Fig. 2, as it didn't make sense. You are correct, we do not show classical or extended allosteric pockets in the figure. This terminology is within the provided references for any reviewer that wishes to follow up.

- Results, lines 160-162: The claim here that V312 forces Y295 to adopt a different conformation is not obvious from your figure. I can see that there's a difference in Y295 at human vs rat and mouse orthologs, but V312 appears to be far away from Y295 and on the basis of the figure alone, doesn't support this particular claim. Could you elaborate on this, perhaps with a clearer figure. This claim is also stated again in the figure legend.

- This was an excellent suggestion. We added Supplementary Fig. 7 to highlight this claim and more clearly show how V312 affects the rotamer of Y295 in the human P2X7R.

- Lines 243-246. An overlay of these structures would be nice to see in the extended data with ortholog specific differences highlighted.

- Thank you for the suggestion. We have now added Supplementary Fig. 10 to highlight the different poses between AZD9056 in the rP2X7R to UB-ALT-P30 in the hP2X7R.

- Lines 282-283: "although no other singular polycyclic hydrocarbon has the success rate of adamantane in medicinal chemistry", success rate of what exactly? Could you please clarify.

- Sorry for the confusion. With this sentence we wanted to highlight that, although several polycyclic hydrocarbons have been explored in medicinal chemistry campaigns, adamantane, is, by far, the more successful scaffold, with eight clinically approved drugs containing adamantane (amantadine, adapalene, arterolane, peficitinib, rimantadine, saxagliptin, vildagliptin, tromantadine). There are a few investigational drugs featuring scaffolds derived from other polycyclic hydrocarbons but, with the exception of the antiviral tecovirimat, they have not reached approval yet. We now rephrase this in the revised manuscript to make it clearer (Lines 287-292).

- Extended data figure 8 legend: No need to include "embedded in POPC bilayers", you've already written that in the methods and it's superfluous here.

- We have removed the text. Thank you.

- Line 666: What is the "f" in f-hP2XR. I couldn't see where in the manuscript that was defined.

- We apologize for the confusion here. The term "f-hP2X7" is an internal nomenclature to reflect the fact that the full protein was used including the ballast domain for the MD simulations. We have now corrected this to say "full-length wild-type hP2X7R" in the revised manuscript, both in the Main Text and in the Supplementary Information.

- Line 675: "ions" are specified but not their species or specific concentration.

- We had this information in the detailed MD simulations protocol in the original version of the manuscript, i.e., that ions were added as NaCl (0.150 M). We have now also added this in the main text in the Methods Section in the revised manuscript (Lines 767).

- Lines 675: How long was your equilibration phase for?

- All the details of the MD simulation protocol used can be found at the end of Supplementary Information after the ligands synthesis details.

- Line 678: “a time step of 1fs and time of 2fs” doesn’t make sense. What was the time step of your simulation?

- The time step was 2 fs along with the SHAKE algorithm in the MD simulations and 1 fs without SHAKE algorithm in the TI/MD simulations. We corrected and clarified this in the revised manuscript in the Methods Section in the main text and in the Supplementary Information where the detailed protocol is described (Lines 796-800).

- Berendsen barostat has been specified for NPT, but what thermostat was used? Was it also Langevin?

- For both NPT and NVT steps the Langevin thermostat was used. We clarified it in the revised manuscript both in the Methods Section in the main text and the Supplementary Information where the detailed protocol is described.

- Line 689: “A known MD simulation protocol was applied”. Please explicitly describe the methods used in the supplementary.

- All the details of the MD simulations protocol are included in the Supplementary Information. The supplementary MD methods were at the very end of the Supplementary Materials file. We are sorry that they were buried. Hopefully you can now fully read the MD methods (lines 1121-1311).

- Line 703: I can’t see this in the supplementary.

- All the details of the MD simulations protocol are included in the Supplementary Information (lines 1121-1311).

- Supplementary figure 6 is not referenced in the text.

- Thank you for catching this! Supplementary Fig. 6 (now Supplementary Fig. 18) is referenced in the main text (lines 301 and 789) and Supplementary methods.

- Lines 761-762 (figure 1): The GDP and Zn²⁺ ions are not shown in Extended data figure 1 as referenced here. Extended Data Fig. 1 shows the 2D chemical structures of P2X7R antagonists.

- Apologies. We meant Supplementary Fig. 2A (not Supplementary Fig. 1) to highlight the location of the ballast. We fixed the callout in the text.

- It is unclear to me why BzATP-induced calcium influx was normalized to an EC80, and to just one of the species? Why is hP2X7R normalized to this value in figure 3? Are these response curves properly normalized in figure 4?

- We thank the Reviewer for catching this error in terminology! We were completely wrong in the way we phrased it. We did not normalize to an EC₈₀. We used an EC₈₀ concentration of BzATP to activate each subtype. For each ortholog/subtype for a given trace, the data were normalized to the calcium signal induced by the respective EC₈₀ concentration of BzATP (when no antagonist was added). All curves are normalized correctly. This is now properly stated in the legends for Fig. 3 and Fig. 4.

REVIEWER COMMENTS

Reviewer #4 (Remarks to the Author):

The authors collected over 8000 Images, processed them in cryoSPARC, and have good resolutions to fit most of the structure well. The processing is standard. Overall, the structures are well determined.

Thank you for kind words on the quality of the structures.

Question:

Is the magnification number for ATP hP2X7R correct? 37 seems wrong.

Thank you for asking about this. A magnification of 37,000x is, in fact, correct. We did not use an energy filter for this data collection, which is stated in the methods section. On the microscope we used to collect this data set, a 37,000x magnification corresponds to a physical pixel size of 0.623 Å/pixel. These numbers are correctly listed in Supplementary Table 1.

Suggestions:

I would have liked to see Supplementary Figure 7 with densities to show that the changes in conformation observed are well determined in the map. The same is true for Supplementary Figure 21—for cholesterol.

Thank you for these excellent suggestions. We have now updated Supplementary Figure 7 to include the cryo-EM densities for the side chains to reveal the conformational changes discussed in the figure. We think the quality of the Cryo-EM density now makes this figure outstanding.

We show the cryo-EM density for the cholesterol molecules in human P2X7 receptor in main text Figure 1B. As discussed, there is no equivalent density for cholesterol in mouse P2X7 receptor or rat P2X7 receptor and thus, there is not really anything to show for those structures.

The general comment is that the allosteric binding site is very similar to that observed in P2X3. While the focus of the paper is on P2X7 - these are very similar in overall structures which is what leads to all the cross reactivities and therapeutic difficulties.

We thank the reviewer for this comment. There are differences between the allosteric pocket of human P2X7 receptor and human P2X3 receptor. As we highlight in the manuscript, the ligand we developed (UB-MBX-46) is much more selective for human P2X7 receptor than for human P2X3 receptor.